# GSCV: Compressing Gaussian Splatting Sequence with Video Codec

## Abstract

This paper presents a novel effective Gaussian Splatting (GS) sequence Compression method that utilizes the Video codec (GSCV). Existing video-based GS sequence compression relies on the Parallel Linear Assignment Sorting (PLAS) to convert GS into smooth 2D maps. Using the vanilla PLAS, however, can generate images exhibiting weak inter-frame correlation, due to its stochastic nature. GSCV incorporates a simple yet efficient Inter-PLAS method to produce close images between the I- and P-frames of GS, enhancing the inter-frame performance of video codec greatly. GSCV also realizes a new pipeline based on the state-of-the-art video codecs with high bit-depth GS images, achieving higher compressibility while simultaneously providing a higher quality upper bound. Experimental results show that the proposed GSCV exhibits obviously improved performance over MPEG video and point cloud-based anchors in GS sequence compression. The code will be released soon.

## 1 Introduction

3D Gaussian splatting (GS) (Kerbl et al., 2023) has greatly advanced novel view synthesis thanks to its fast generation speed and impressive quality. However, the explicit GS primitive format induces a large data volume, which attracts considerable attention for effective GS data compression (Bagdasarian et al., 2025).

To facilitate GS compression standardization, the Moving Picture Experts Group (MPEG) has established a GS compression Ad Hoc group and delineated two lines of investigation: the A-3DGS and I-3DGS tracks (WG7, 2025). A-3DGS uses 2D videos/images as the input. With any agreed-upon GS-related representation as the intermediate, A-3DGS involves training GS models as part of the encoding process (i.e., optimization-based compression). I-3DGS aims to compress the trained GS data (i.e., optimization-free compression), resulting in a symmetric output with the same format as the input. At present, most GS compression efforts focus on A-3DGS. They include HAC (Chen et al., 2024a) and CompGS (Liu et al., 2024). Using an effective context model and rate-distortion (RD) loss function, they can achieve a compression ratio of more than 60 to 100x without noticeable distortion (Xing et al., 2025). However, these optimization-based algorithms typically involve a complex coding process and rigid rate control (Yang et al., 2025) based on GPU, which restricts their application to certain use cases. The lightweight requirement from the industry (WG4, 2025c) promotes more attention devoted to I-3DGS. One promising approach to develop lightweight solutions is to compress GS data using canonical codecs such as the video or point cloud (PC) codec (Sullivan et al., 2012; Schwarz et al., 2019), where the compression is mainly performed on CPUs.

Video-based GS compression depends on the use of Parallel Linear Assignment Sorting (PLAS) (Morgenstern et al., 2024) to convert GS data to a group of smooth 2D maps. PLAS is a progressive process inspired by image sorting (Barthel et al., 2023). It first randomly puts the primitive attributes into 2D grids, then gradually generates a smoother image via Gaussian blurring from being coarse to fine as a target to sort the pixels until the 2D grids converge to a smooth distribution. For intra-frame compression strategies, PLAS significantly improves the video codec performance, reporting 6 to 20x compression ratios on HEVC (Do et al., 2025). Based on the preliminary MPEG experiments, for tracked GS sequence compression (see Section 6 for an explanation of "tracked"), the video-based anchor (GSCodec Studio (Li et al., 2025)) is superior to the PC-based anchor (GPCC v1 (WG7, 2023)) owing to its efficient inter-frame compression (WG4, 2025b). However, few works

focused on GS inter-frame compression based on video codecs, and the current anchor has obvious limitations, which motivate the study of this paper.

To the best of our knowledge, GSCodec Studio pioneered the open source solution that supports GS inter-frame compression based on video codecs. In its inter-prediction mode, PLAS is used for the first frame (i.e., the I-frame), and then the I-frame PLAS index is utilized for sorting the following frames (i.e., the P-frames) within the same group of pictures (GoP) to generate videos. This operation is built upon two assumptions: 1) the arrangement of the I- and P-frame primitives demonstrates an explicit spatial correspondence, whereby primitives with identical indices constitute spatial nearest neighbors sharing close attributes; and 2) directly using the I-frame PLAS index can generate smooth and close P-frame images.

Nevertheless, the above assumptions can be violated in practical applications, resulting in suboptimal results as shown in Fig. 1. For the first assumption, GS data is characterized by the permutation invariance as PC (Li et al., 2018). Even though the initial GS data might share some correspondence between adjacent frames (e.g., they may be generated from the same canonical reference after fine-tuning), a considerable number of preprocessing operations before compression, such as sampling and editing (Hanson et al., 2025; Chen et al., 2024b), can scramble the primitive order. Even a simple primitive shuffle operation, which does not change GS data and rendering results, can incur obvious performance degradation (see Section A.7). For the second assumption, GS data is not injective due to $\alpha$-blending (Tewari et al., 2022), which means spatially neighboring primitives in the I- and P-frames may inherently possess substantially different attributes, especially in the large motion region. This observation implies that index-based primitive matching alone is insufficient to ensure that the P-frame images are smooth and close to the corresponding I-frame images. Besides, the current video-based anchor uses 16 bit depth (BD) for coordinates and 8 BD for quantising other attributes, which is not enough for complex GS scenes. Results from (Zaghetto et al., 2024) showed that the optimal coordinate quantization BD lies within the range of 14 to 18, depending on particular datasets. Considering that GS is very sensitive to coordinate distortion, it would be better to use a larger BD during quantization and realize quality-bitstream balance via compression. Otherwise, the quality upper bound will be limited even under lossless compression.

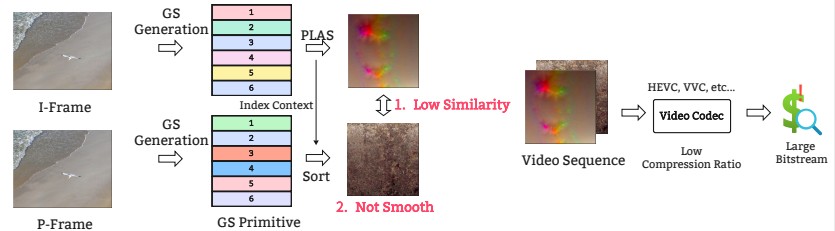

Figure 1: Examples of current inter-frame GS coding based on the video codecs.

To improve the GS sequence compression performance and robustness, we propose **GSCV**, a new pipeline for **GS** sequence **C**ompression based on **V**ideo codec. First, an effective and simple Inter-PLAS method, which consists of two steps, is proposed to reduce the PLAS image difference between adjacent GS frames. Inter-PLAS adopts a spatial-stable initialization (SSI) module designed to address the GS permutation-invariance issue, leading to stable I-frame PLAS image generation without affecting the compression efficiency. It also enables initializing the P-frame with low complexity using the I-frame context. Additionally, Inter-PLAS has an anchor-based PLAS refinement, which is proposed to optimize the P-frame image for better inter-frame prediction. Second, we establish a new PLAS image compression framework compatible with the state-of-the-art (SOTA) video codecs, including HEVC and VVC (Bross et al., 2021). For a higher quality upper bound, 10 BD images are used for video sequence generation after Inter-PLAS. The coordinate maps, which are quantized by 20 BD, are split into two 10 BD images. Channel padding is adopted for attributes (e.g., opacity and rotation) that are not integer multiples of three channels to simplify compression configuration, as well as making it compatible with different image or video codecs. GSCV reports superiority on both MPEG tracked and semi-tracked datasets over benchmark methods.

Our contributions can be summarized as follows:

• A new GS sequence compression pipeline based on video codecs, GSCV, is proposed to take advantage of advanced video codecs for efficient inter-frame GS sequence compression;

• We propose a simple yet effective Inter-PLAS scheme to generate close 2D maps for adjacent GS frames to improve the inter-frame prediction performance. We use high BD images to form video sequences, which can provide wider bitrate and quality ranges;

• The experiment results show that the proposed GSCV is superior to MPEG video- and PC-based anchors. Ablation study demonstrates the robustness and generalization capability of GSCV.

## 2 RELATED WORK

GS compression has witnessed impressive progress in the past two years. As introduced in Section 1, GS compression can be divided into two tracks according to the MPEG community: A-3DGS and I-3DGS. We shall provide a summary in this section.

**A-3DGS**: A-3DGS optimizes the GS generation process by adding extra constraints during training to obtain a more compact data representation. Scaffold-GS (Lu et al., 2024) proposed an anchor-based latent representation for primitive description, which reduces the spatial redundancy and also serves as the baseline of HAC (Chen et al., 2024a) and HAC++ (Chen et al., 2025b). CompGS (Liu et al., 2024) also used an anchor-based approach with an entropy model to realize efficient adjacent primitive feature prediction. Another prevalent compact representation is using a codebook and vector quantization to map the GS attributes to a limited feature space, such as (Navaneet et al., 2025; Niedermayr et al., 2024). Considering that the GS densification process can generate many redundant primitives, LightGaussian (Fan et al., 2024) and EAGLES (Girish et al., 2025) developed effective GS pruning methods by calculating an importance score for each primitive. Empirical results show that removing 50% to 60% of primitives followed by finetuning can still offer a quality comparable to that of the vanilla GS. In summary, A-3DGS methods can realize an impressive compression ratio without evident quality degradation, indicating there is a huge room to optimize the GS generation process, specifically over the densification and high-dimensional spherical harmonic (SH) coefficients. These advances also inspired the study of the I-3DGS approaches.

**I-3DGS**: Different from A-3DGS, I-3DGS methods focus on how to compress trained GS data without resorting to training optimization that relies on the GPUs. GS data format is similar to that of PC. Therefore, an adaptive voxelization was designed in (Wang et al., 2025) for GS. It takes into consideration that the primitive distribution is irregular to facilitate the usage of the MPEG GPCC codec on GS compression. Also inspired by the PC compression (Shao et al., 2017), GGSC (Yang et al., 2024) used a graph signal processing (GSP)-based approach to compress GS attributes through discarding high-frequency components. HGSC (Huang et al., 2025) proposed an optimization-free GS pruning method based on LightGaussian, followed by a hierarchical strategy to divide primitives into different layers for progressive coding. FCGS (Chen et al., 2025a) used grids to establish an effective context model for learning-based GS compression. GSCodec Studio (Li et al., 2025) is the first modular framework that supports GS reconstruction, compression, and rendering, in which video- and PC-based GS compression are unified under this pipeline. Except for GSCodec Studio, other methods reviewed in this section all focus on GS intra-frame compression, which motivated us to further explore how to realize effective and stable GS inter-frame compression in this paper.

## 3 BRIEF REVIEW OF PLAS

PLAS is a heuristic algorithm to sort and map GS attributes into 2D maps. Ideally, the generated 2D maps should be as smooth as possible so that they can be effectively compressed using video codecs. Given a GS data $G \in R^{n \times 59}$, where $n$ represents the number of primitives in GS and 59 is the number of feature channels, including 3, 3, 45, 1, 3, and 4 channels for 3D coordinates, color direct current (DC), color SH, opacity, scaling, and rotation. A simple pruning, which is based on the linear processing of scaling and opacity, is first applied so that after discarding a minimal set of GS, the remaining primitives can be accommodated in a square 2D grid. Next, a **random sorting** is used to map the primitive attributes onto 2D grids. $m$ ($m <= 59$) images are obtained as input for the downstream operations, where $m$ corresponds to the number of attributes selected as references for smooth evaluation. For the pixel values that are from different images but have the same position, they originate from the same primitive, indicating that no extra signaling is needed for attribute alignment after sorting.

PLAS takes a progressive approach: 1) a set of Gaussian blur window radius sizes $r$ is selected based on the resolution of the image $I \in R^{H \times W \times m}$. $r = \{r_i, \tau\}$ with $r_1 = max(H, W)/2 - 1$, and $r_i = r_{i-1} \times \tau$. $\tau \in (0, 1)$ is a decay factor, and the minimum value of $r_i$ is set to be 1; 2) a set of block sizes $B_i$ is calculated in accordance with the radius $r_i$. Specifically, $B_i = max(B_{min}, floor(r_i \times 2 + 1) + 1)$. $B_{min}$ is the minimum block size which is set to 16 in the vanilla PLAS. The definition of $B_i$ ensures that the pixels in the blocks can be divided into groups, each of which has 4 elements; 3) for iteration $i$, a target smooth image $T_i$ is generated from $I_{i-1}$. $T_i$ is in fact the Gaussian blurred version of $I_{i-1}$ with the kernel size $\sigma_i = floor(r_i \times 2 + 1)$, i.e., $T_i = blur(I_{i-1}, \sigma_i)$. The iteration starts with $I_0 = I$, the original image; 4) both $T_i$ and $I_{i-1}$ are divided into sub-images with size equal to the block size $B_i$ for batch operation. PLAS permutes the pixels of the image from the previous iteration $I_{i-1}$ within each block, making them as close as possible to those of the corresponding block in $T_i$. Note that the permutation is carried out for a group of 4 pixels selected randomly from $I_{i-1}$, leading to $4! = 24$ possible permutations. The one with the smallest L2 loss is chosen to produce $I_i$, the image to be passed on to the next iteration; 5) given that $\sigma_i$ and block size $B_i$ are monotonically decreasing with respect to the iteration index $i$, a coarse-to-fine permutation is applied for the original image $I$ and results in the final PLAS image with smooth texture.

## 4 INTER-PLAS IMAGE GENERATION

Due to the use of random initialization and permutation, the vanilla PLAS generates different images for the same or close GS content, as shown in Fig. 2. These images are suitable for the intra-compression mode of the video codec but are challenging for inter-prediction. This motivates us to

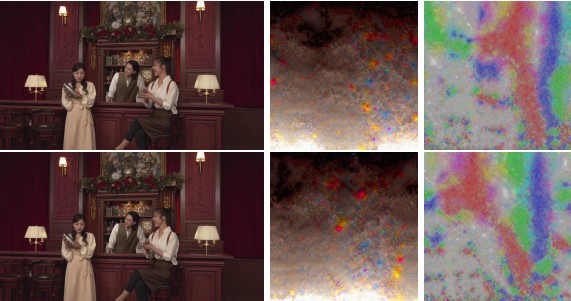

Figure 2: PLAS results of "bartender" frames 1 and 2. From left to right: the GS rendering image, color DC PLAS image, and scale PLAS image.

design the new effective algorithm, Inter-PLAS. It has improved inter-prediction using two steps, namely SSI and anchor-based PLAS refinement. The diagram of Inter-PLAS is given in Fig. 3.

### 4.1 SPATIAL-STABLE INITIALIZATION (SSI)

PLAS applies random sorting as the initialization step, which can produce different results for even the same GS data. Even though we may fix the random seed, the permutation-invariant characteristic of GS data can still lead to different results, indicating current PLAS is non-deterministic and unstable. For GS sequence coding, a stable, simple, and effective initialization method is required for generating highly reproducible results. To simplify the problem, we use the I- and P-frames from video coding as the reference and target GS frames.

A reasonable assumption for the initialization process is that the spatially proximate primitives should have close attributes. This indicates that for each primitive from the P-frame, we can match it to the nearest neighbor from the I-frame and make them share the same image position. However, this one-to-one matching requires using Earth Mover's Distance (EMD) whose complexity is $O(n^3)$ (Kuhn, 1955). For GS data that generally has hundreds of thousands to tens of millions of primitives, it is extremely expensive to evaluate the EMD (Yang et al., 2023). Considering that we only need a stable initialization rather than requiring this initialization to generate similar images for the I- and P-frames, we address the spatial matching problem with spatial sorting based on Morton code (Morton, 1966).

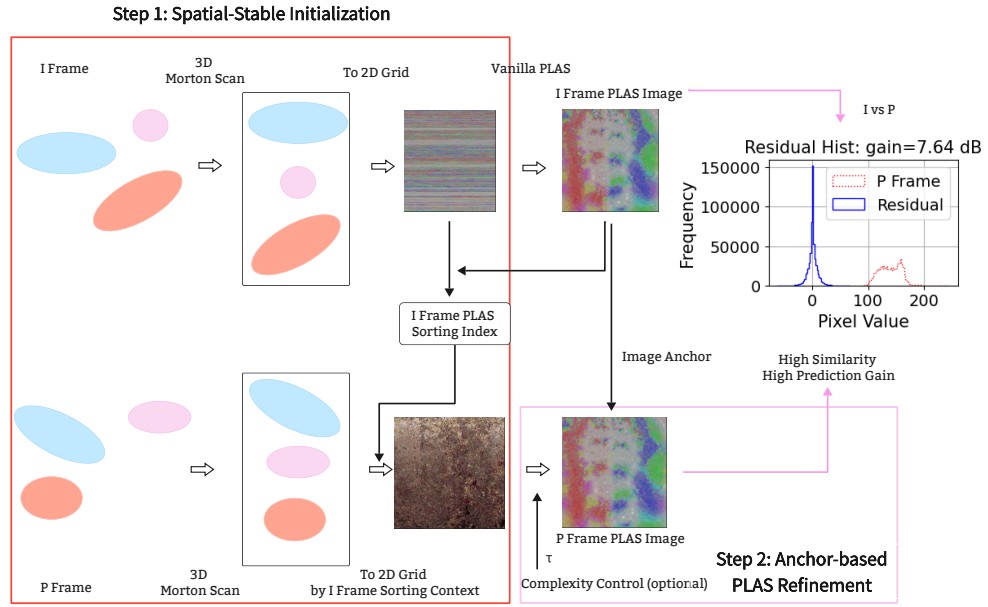

Figure 3: Framework of Inter-PLAS.

Morton code interleaves the binary representation of the 3D coordinates of the primitive. A smaller Z value reflects that the primitive is closer to the "initial position," which is set to be the top-left corner or the original position. For the I- and P-frames, we first sort the primitives based on their Morton code. The primitives with the same sorting index from the I- and P-frames are roughly considered proximal to each other in the EMD sense. Morton code sorting is lightweight, e.g., it only requires around 0.1 seconds for GS with half a million primitives. Empirical results show that, compared with random initialization, using the Morton code-based sorting as initialization and then performing the vanilla PLAS can generate more similar images for the I and P-frames. Close I-frame all-intra compression performance can also be obtained (see Section 6.3.1), indicating that PLAS is robust to the initialization method in terms of all-intra compression.

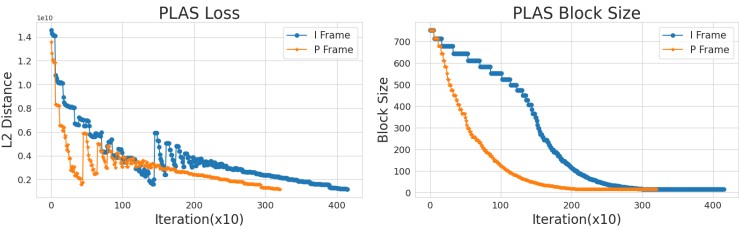

Figure 4: PLAS loss and block size variation curve of "bartender" frame 1 (I-frame) and 2 (P-frame).

For the I-frame, we produce a 2D map $I_{ini}$ in a row-by-row manner (see section 6.3.1 for other manners), with the row indices coming from the Morton code sorting results. Then, a vanilla PLAS is applied to obtain the attribute image $I_{PLAS}$. The I-frame PLAS sorting index $Idx_{PLAS}$ is saved as the context for the P-frame initialization. For the P-frame, after 3D Morton code sorting to obtain $P_{ini}$, $Idx_{PLAS}$ is used to re-arrange the index of the $P_{ini}$ again and generate the 2D maps $P'_{ini}$ as the initialized version. $P'_{ini}$, a smoother version of $P_{ini}$, can be fed into the vanilla PLAS to obtain $P_{PLAS}$, which saves around 15% GPU time. The benefit of this design is illustrated using the MPEG tracked dataset "bartender": performing PLAS on $I_{ini}$ takes 36s, while 30s for $P'_{ini}$. Based on the PLAS distance and block size variation curves shown in Fig. 4, we found that PLAS exhibits obviously faster convergence speed, especially for the large block size stage. For the I-frame, PLAS uses around 4,000 iterations to converge to the final results, while the P-frame only requires 3,000 iterations. This implies that a smoother initialization can reduce the time complexity of PLAS. Considering that the P-frame requires fewer iterations, especially for larger block sizes, we can use a smaller $\tau$ to accelerate block size convergence while ensuring satisfactory results.

## 4.2 ANCHOR-BASED PLAS REFINEMENT

Although $P_{PLAS}$ generated using SSI and the vanilla PLAS is smooth in a whole, and it is closer to $I_{PLAS}$ than applying the vanilla PLAS on the I- and P-frames separately, evident discrepancies may continue to be present in some content or attributes. An example is shown in the first and second rows of Fig. 5. Using a video codec such as HEVC in the inter-prediction mode only has around $8\% \sim 14\%$ bitrate savings compared to the all-intra mode. This motivates us to propose an anchor-based PLAS refinement to adjust $P'_{ini}$ rather than directly using a vanilla PLAS. Specifically, we disable the generation of the blurred target $T_i$ in PLAS and use $I_{PLAS}$ instead as the anchor to progressively smooth $P'_{ini}$, resulting in the proposed anchor-based PLAS refinement. Fig. 5 third row shows the results of applying the modified PLAS with the proposed anchor-based refinement to $P'_{ini}$. A prediction gain (PG) is used to quantify the benefits of inter-frame compression:

$$\mathrm{PG(I,P)} = 10\log_{10}(\mathrm{E_I/E_{res}}),\ \mathrm{E_I} = \mathrm{var(I)},\ \mathrm{E_{res}} = \mathrm{var(I-P)}, \tag{1}$$

where $\mathrm{var}(\cdot)$ represents the variation operator. For the vanilla GS, only the color DC map has 6.61 dB PG, while scaling and rotation have negative PG, which means: 1) the pixel-to-pixel similarity between the I- and P-frames is low; and 2) the inter-frame spatial correlation of color DC is more pronounced. An obviously increased PG is observed for the anchor-based PLAS refinement, which means I- and P-frames are now closer to each other, suggesting better inter-prediction potential.

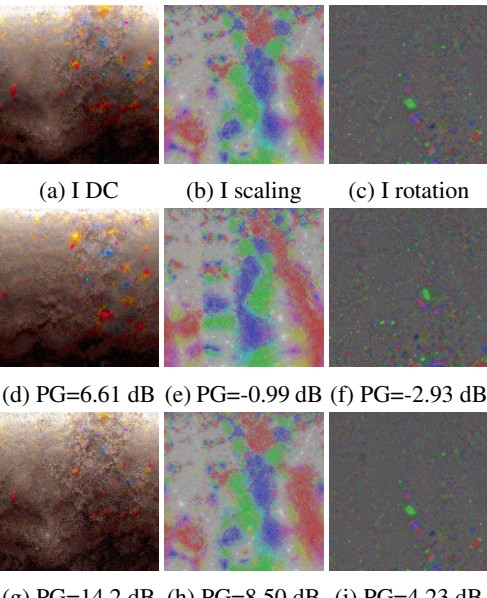

(a) I DC      (b) I scaling      (c) I rotation

(d) PG=6.61 dB   (e) PG=-0.99 dB   (f) PG=-2.93 dB

(g) PG=14.2 dB   (h) PG=8.50 dB   (i) PG=4.23 dB

Figure 5: Anchor-based PLAS refinement. First row: PLAS results for the 1st frame of "bartender"; second row: PLAS results for $P'_{ini}$; third row: the proposed anchor-based PLAS refinement for $P'_{ini}$.

The vanilla PLAS and proposed anchor-based PLAS focus on different perspectives, namely intra-frame smoothness v.s. inter-frame similarity. Based on Fig. 5 (g), high-frequency noise like "salt&pepper" noise is observed, while Fig. 5 (d) is spatially smoother. For the video codecs in the inter-prediction mode, both intra-frame content smoothness and inter-frame similarity will influence the P-frame compression efficiency. With the developed Inter-PLAS, higher inter-frame similarity is achieved: over 20% bitrate saving can be obtained for P-frame coding.

One potential shortcoming of the anchor-based PLAS refinement comes from directly using a smooth image as the target, which is inconsistent with PLAS having a coarse-to-fine process, leading to slow convergence. Fortunately, inspired by the discussion about Fig. 4, we find a very easy method to realize complexity control without sacrificing the compression efficiency significantly, which is reported in Section. A.8.

## 5 GAUSSIAN SPLATTING SEQUENCE CODING USING VIDEO CODEC

Based on the images generated by Inter-PLAS, the proposed effective GS sequence compression scheme GSCV is established, as shown in Fig. 6. It can be seen that first, we combine 8 frames into 1 GoP, with the first frame designated as the I-frame and the rest 7 frames as the P-frames. SSI + Vanilla PLAS and Inter-PLAS are used for the I- and P-frames respectively to project primitives onto 2D grids. After that, 20 BD is used to quantize the coordinates and 10 BD is applied to quantize other attributes. For each GS frame, we generate 1, 15, and 1 frame images for the color DC, SH, and scaling. For opacity and rotation, we use the mid-value of BD to pad two channels to form 1 and 2 images, respectively. Most video codecs cannot compress 20 BD images directly. To improve the compatibility of GSCV, we split the coordinates map into 2 images, namely the high part (HP) and low part (LP) of data. Each part takes 10 BD.

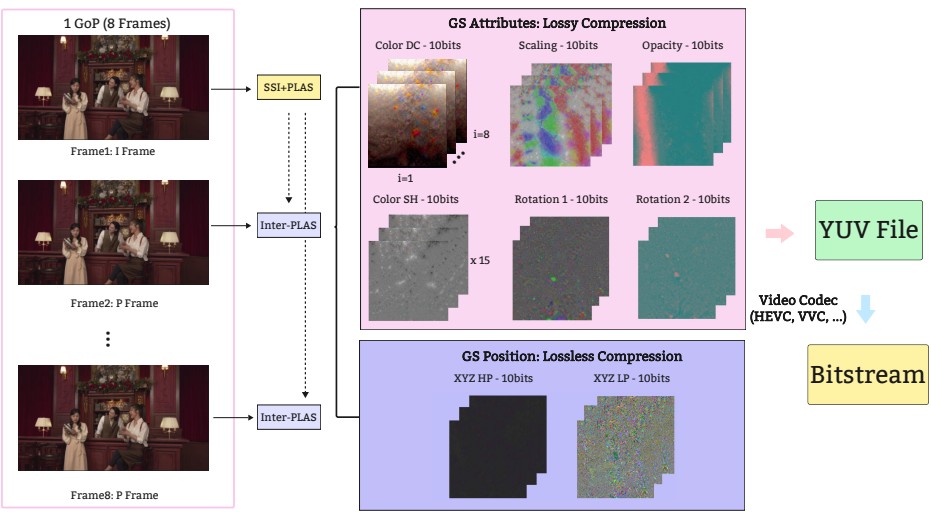

Figure 6: Diagram of the proposed GSCV.

For each GS frame, 22 images are generated. We extract the corresponding images from each frame within a GoP to group 22 videos and save these videos in YUV444 format. Lossless compression is used for coordinates, while other attributes employ the lossy mode. Note that we directly consider the video data as YUV rather than RGB, omitting the operation that converts data from RGB to the YUV color space for YUV file generation. The reasons are twofold: 1) except for the color DC and SH, there is no inter-relationship analogous to the other attributes; applying an RGB-to-YUV transformation is thus redundant and offers no benefit for compression; 2) even with a lossless compression, the RGB-to-YUV transformation introduces inevitable calculation errors that result in noticeable quality degradation (especially for the coordinate HP image) at high bitrate due to the lossy forward and inverse processes. The influence of introducing RGB-to-YUV is reported in Section A.6.

## 6 EXPERIMENTS

In this section, the performance of the proposed GSCV on an authoritative dataset from MPEG is reported. We compare it against two anchors adopted in the MPEG GS compression working group, namely GSCodec Studio and GPCC v1 (see Section A.2 for an introduction). The ablation study is presented at the end of this section to demonstrate the effectiveness of different modules.

### 6.1 DATASET AND GSCV IMPLEMENTATION

Three GS sequences with **tracked** and **semi-tracked** versions from the MPEG GS compression working group are used to test the proposed method: "bartender", "cinema", and "breakfast" (see Section A.3 for details). Each sequence has 32 frames and we use the first 8 frames as 1 GoP in the following experiment. Here, "tracked" means that 32 frames GS are refined from a shared canonical

GS model, which was trained using the merged point cloud with ground truth images from all viewpoints and time instants. Densification is disabled during the refinement process, ensuring that all the frames have the same number of primitives. "Semi-tracked" means densification is allowed during the refinement, which results in a different number of primitives for different frames. More details about the tracked and semi-tracked sequences generation can be found in (Jeong et al., 2025). To simulate the practically interesting situation that some operations might affect the permutation of primitives, we **shuffle** randomly the tracked data of each frame before testing. For video sequence generation, we use LightGaussian to **prune** the semi-tracked dataset to ensure frames within 1 GoP have the same number of primitives by removing the minimum element. **No shuffle** is introduced for semi-tracked data. The implementation of the proposed Inter-PLAS is based on the vanilla PLAS (HHI, 2024) realized in PyTorch. All the attributes are used in PLAS image generation, i.e., $m = 59$ (see Section A.10 for other attribute settings). Two popular video codecs, HEVC reference software (HM) version 18.0 (H.265) and VVC reference software (VTM) version 23.11 (H.266), are used as the codec anchor of GSCV. The detailed codec configuration is presented in Section A.1. As suggested by (WG4, 2025a), we use views 9 and 11 for "bartender" and "cinema", while views 6 and 8 for "breakfast" to compute the quality metrics. The reported results are obtained on a single NVIDIA L40s GPU and AMD EPYC 7H12 CPU.

## 6.2 OVERALL EVALUATION

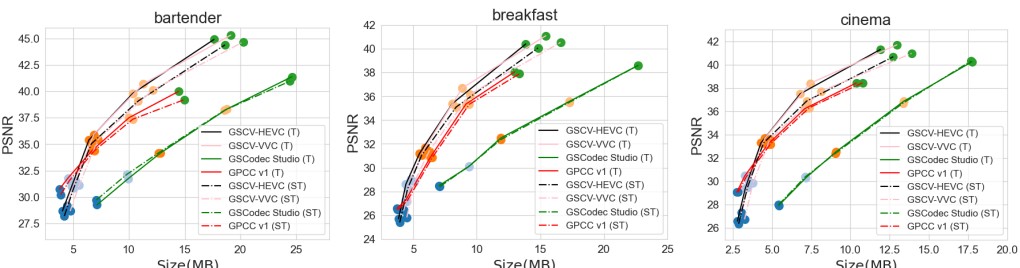

Figure 7: Performance comparison on MPEG dataset. "T" and "ST" mean tracked and semi-tracked.

Fig. 7 illustrates the rate (MB) vs. distortion (RD) curves of the proposed GSCV on the MPEG tracked and semi-tracked dataset. The results of rate (bits per primitive (BPP)) are shown in Section A.4, and the RD curves for per-components are shown in Section A.14. The quantization parameters (QPs) of five bitrates are presented in Section A.1. GSCV shows obvious superiority over GSCodec Studio and GPCC v1. For example, although both use video codecs, GSCV reports around 8 dB gain on "bartender" compared with GSCodec Studio when the bitstream is 10MB, thanks to the proposed Inter-PLAS and new compression framework. The inter-frame compression performance of GSCodec Studio highly relies on the initial primitive permutation between different frames as discussed in Section A.7. With VVC, close performance compared with HEVC is obtained. This is not consistent with the case of ordinary videos where VVC can achieve 50% gain over HEVC. The reason is that the current video codecs are optimized for natural images that are completely different from the PLAS images. Based on the residual analysis in Section A.13, the PLAS image shows a more pronounced heavy-tail behavior than natural images. The same phenomenon also occurs when using FFMPEG libx265 and libx264, as reported in Section A.16. This also demonstrates that the superiority of the proposed GSCV does not mainly come from using a better video codec than GSCodec Studio. Instead, it originates from the more effective and robust pipeline pivoting a better GS video generation.

Second, GSCV provides better performance than GPCC v1 with the middle to high bitstreams, while poorer results with the low bitstream. The underlying reasons are twofold: 1) the setting of the color DC QP is too large, leading to inappropriate bitrate distribution. We tried other QP and YUV modes in Section A.6, and 2) GPCC v1 is more effective in encoding the coordinate which is always lossless compressed. A greater proportion of geometry at lower bitrates can lead to improved GPCC performance. A detailed bitrate allocation and coding time are reported in Section A.5 and A.16 with sample snapshots given in Section A.20. PALS coordinate LP images are close to noise maps, resulting in a limited performance when using video codecs. To solve this problem, we propose an optimized coordinate compression in Section A.15, in which we found that using 12 BD for HP and

8 BD for LP, as well as using Lempel–Ziv–Markov chain Algorithm (LZMA) (Ziv & Lempel, 2003) to compress LP images can realize a higher compression ratio with faster coding speed.

Third, GSCV reports consistent performance on both tracked and semi-tracked sequences, indicating the robustness of the proposed Inter-PLAS. Semi-tracked sequences are more challenging than the tracked version, indicating densification tends to amplify the inter-frame differences.

## 6.3 ABLATION STUDY

### 6.3.1 INFLUENCE OF PLAS INITIALIZATION ON IMAGE GENERATION AND COMPRESSION

The first step of Inter-PLAS provides a good initialization method to solve the problem introduced by GS permutation-invariant characteristics, generating a stable I-frame image as the P-frame's context. The initialization should not damage the intra-frame compression effectiveness of the I-frame, considering that the I-frame occupies more bitstream than the P-frame. We compare the results of GSCV all-intra compression mode on PLAS image with different initialization methods followed by the vanilla PLAS with tracked sequences: 1) random initialization; 2) Morton1, using 3D Morton scan first and then generating 2D maps row by row; 3) Morton2, using 3D Morton scan first and then mapping it to 2D maps based on 2D Morton scan order; 4) using a PLAS index from another frame as context. The visualization of four initialization methods and RD are shown in Figs. 8 (a) and (b). We can see that the RD curves of different initialization methods are almost the same, indicating that the PLAS is robust to initialization in all-intra compression. However, using another frame as context requires the shortest PLAS time, indicating that a smoother initialization can accelerate PLAS convergence to stable states.

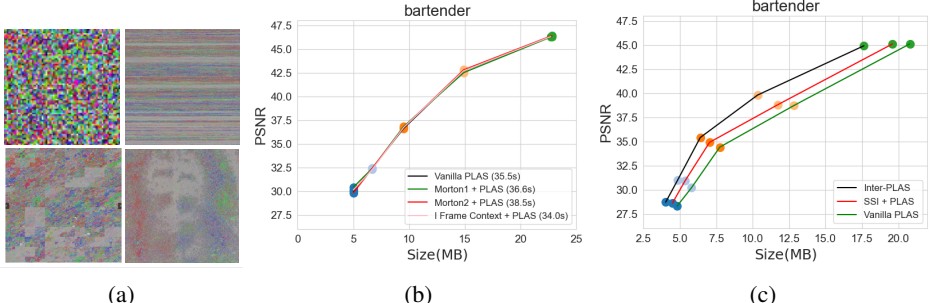

(a)                    (b)                                    (c)

Figure 8: Ablation study on (a)-(b): PLAS initialization and (c): effectiveness of Inter-PLAS.

### 6.3.2 EFFECTIVENESS OF INTER-PLAS

Although GSCodec Studio also supports inter-prediction for GS sequences as GSCV, it directly uses the PLAS index from the I-frame to sort P-frames, which cannot ensure that the P-frame image aligns well with the I-frame. To highlight this problem, we use GSCV based on three different methods to generate images with tracked sequence: 1) using the vanilla PLAS on each frame separately; 2) using SSI to initialize the P-frames followed by the vanilla PLAS to refine images; and 3) the proposed Inter-PLAS that uses the I-frame to refine P-frame images. The results are shown in Fig. 8 (c). As we discussed for Fig. 2, directly using the vanilla PLAS will generate different images for close content due to the random process, resulting in the worst inter-prediction results. Using a good context to initialize the P-frame with PLAS can generate better P-frame images. However, the vanilla PLAS only focuses on intra-frame smoothness, while there is no supervision on inter-frame similarity, leading to limited gain for inter-prediction. The proposed Inter-PLAS solves the image alignment problem by a simple and effective strategy with high PG, showing an obvious gain on inter-prediction cases.

## 7 CONCLUSIONS

In this paper, we propose GSCV, an effective GS sequence compression pipeline based on video codecs. We design an Inter-PLAS method to generate high-correlation GS images, which can facilitate the inter-frame compression for video codecs greatly. We also design a new GS image

compression pipeline based on SOTA video codecs. By using high BD images and being compatible with prevalent video codecs, GSCV reports obviously better performance on both tracked and semi-tracked sequences than video- and PC-based anchors used in the current MPEG standardization study.

**Limitations:** PLAS images are very different from natural images, leading to restricted compression ratios of the proposed GSCV, which is now based on generic video codecs. One potential enhancement to solve this problem is to use a "Sandwich network" (Guleryuz et al., 2024) to convert PLAS images to a data distribution more suitable for traditional video codec compression. Dividing the GS data into multiple video sequences and compressing them separately cannot fully utilize the primitive intra-channel correlations, especially for the color SH (see Section A.9). We will explore how to better group color SH images in future work. For tracked GS data after permutation, Inter-PLAS cannot completely recover the primitive correspondence due to its heuristic mechanism inherited from PLAS.

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

## A  APPENDIX

### A.1  BITRATE SETTING DETAILS

In Table 1, we show the QP values of five bitstreams for GSCV. For HEVC lossy compression, "encoder_randomaccess_main_rext.cfg" is used for inter-frame compression; for lossless compression, two flags are set: "TransquantBypassEnabledFlag = 1" and " CUTransquantBypassFlagForce = 1". For VVC lossy compression, "encoder_randomaccess_vtm.cfg" is used; for lossless compression, "lossless444.cfg" and "lossless.cfg" are applied. We report using HM18.0 "encoder_low_delay" configuration with different coding sequencing in Section A.11.

| Rate | Coordinate | QP | | | | |
| --- | --- | --- | --- | --- | --- | --- |
| | | Color DC | Color SH (degree 1/2/3) | opacity | scale | rotation |
| R05 | | 0 | 0/0/0 | 0 | 0 | 0 |
| R04 | | 7 | 7/12/17 | 7 | 7 | 2 |
| R03 | lossless | 17 | 17/22/27 | 17 | 7 | 2 |
| R02 | | 37 | 22/27/32 | 17 | 12 | 7 |
| R01 | | 47 | 32/37/42 | 22 | 12 | 17 |

Table 1: QP of five bitrates of GSCV

### A.2  BASELINE ANCHORS

**GSCodec Studio**: GSCodec Studio (Li et al., 2025) uses FFMPEG x265 codec, which supports GoP-wise compression with inter-prediction. The default GoP size is 16, and 8 BD is used to quantize the PLAS image. In our experiment, we change the GoP size to 8 for fair comparison.

**GPCC v1**: GPCC v1 (WG7, 2023) was originally designed for PC data compression. MPEG extended its interface that supports GS data loading and used it as the PC-based anchor for GS compression. The compression of coordinates and other attributes inherits the compression method of PC: Octree for coordinates and RAHT (De Queiroz & Chou, 2016) for attributes compression. It uses 18 bits to quantize coordinates and 12 bits for other attributes. Current GPCC v1 does not support GS inter-frame compression, therefore, we use all-intra mode in Section 6.

## A.3 DATASET INFORMATION

The number of primitives for MPEG tracked and semi-tracked sequences is shown in Table 2. For tracked sequences, the primitive number is the same for different frames. For semi-tracked sequences, the red number is the minimum primitives within this GoP, which will be used as the target number after pruning. Considering that densification is allowed during refinement, semi-tracked sequences have more primitives than the tracked version.

| Sequence | | Primitive Number | | | | |
|---|---|---|---|---|---|---|
| | Bartender | | Breakfast | | Cinema | |
| Frame Index | Tracked | Semi-tracked | Tracked | Semi-tracked | Tracked | Semi-tracked |
| 1 | | 570255 | | 531263 | | 426968 |
| 2 | | 570104 | | 530997 | | 426752 |
| 3 | | 569842 | | 530998 | | 426497 |
| 4 | 567724 | 569733 | 528154 | 530857 | 423249 | 426209 |
| 5 | | 569556 | | 530686 | | 426145 |
| 6 | | 569285 | | 530582 | | **425953** |
| 7 | | 569299 | | **530509** | | 426035 |
| 8 | | **569246** | | 530524 | | 426033 |

Table 2: Number of primitives for MPEG tracked and semi-tracked sequences.

## A.4 BPP-BASED RD CURVE

The results of the rate (BPP) vs. distortion are summarized in Fig. 9. The primitive numbers of the tracked and semi-tracked "bartender", "breakfast", and "cinema" are 567,724/569,246, 528,154/530,509, and 423,249/425,953, respectively.

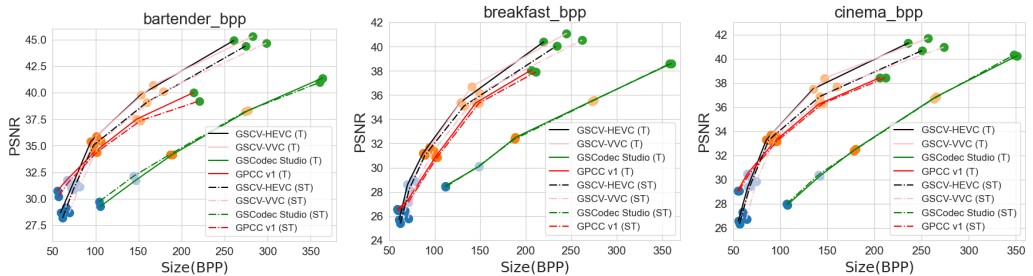

Figure 9: PSNR performance comparison on MPEG tracked and semi-tracked dataset.

## A.5 BITRATE ALLOCATION

To illustrate the bitrate distribution for different components on GSCV, we use tracked "bartender" rate 1, 3, and 5 with HEVC as an example. The BPP is shown in Fig. 10. We can see that for high BPP, color SH occupies more bitstreams, while geometry-related features contribute more to low BPP. Considering that GS quality is more sensitive to geometry-related features, saving bitstream from color-related features can provide a better quality-bitstream tradeoff. The coding time, which is shown in Fig. 11, illustrates that the decoding is faster than encoding, and this coding time is thus only for complexity illustration purpose. Currently, we establish GSCV based on the HM or

VTM software, which is mainly used for standardization studies. The coding speed is relatively slow owing to the lack of engineering optimizations (e.g., using multithreaded processing). Faster coding speed can be achieved by choosing "encoder_lowdelay_main_rext.cfg" file or using optimized codec platforms (e.g., FFMPEG or VVdeC (Wieckowski et al.))

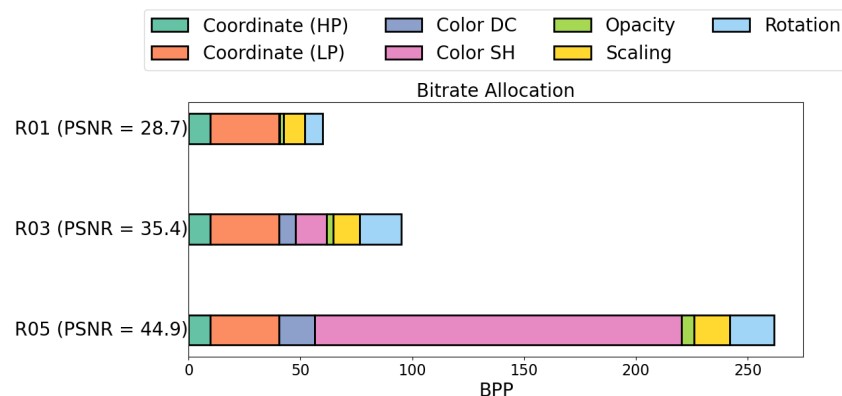

Figure 10: Bitrate allocation of GSCV.

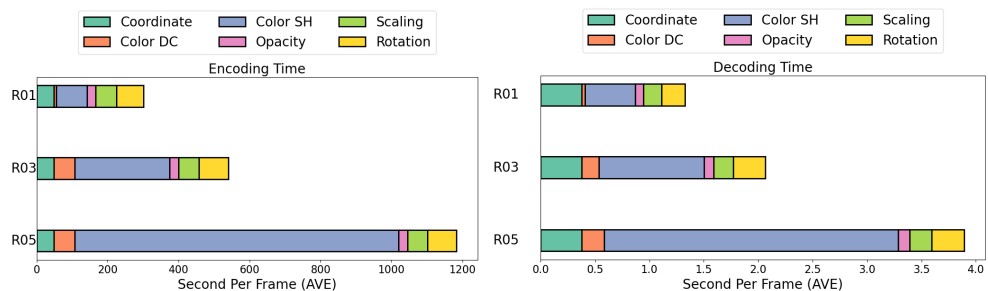

Figure 11: Coding time of GSCV.

### A.6    BITRATE REDISTRIBUTION

In Fig. 7, we find that the proposed GSCV exhibits faster PSNR decay since the third bitrate point. The reason may be that we use a large QP for color DC, while color DC serves as a cornerstone for visual quality. Therefore, we try the other QP setting (noted as C2, while the QP reported in Table 1 is referred to as C1): for R01 to R03, we use QP = 7 for color DC as suggested by (Do et al., 2025). Besides, color SH contributes to the majority of the overall bitstream as shown in Fig. 10. We use YUV420 to reduce the bits of chrominance from color SH, and use C1 and C2 to generate new RD curves. The results are shown in Fig. 15. We find that: 1) C2 yields better results than C1 at low to medium bitrates, indicating that more bitrates should be devoted to the color DC component; 2) using YUV420 for color SH demonstrates even better performance at low to medium bitrates, while the quality ceiling has been substantially decreased even though QP=0. It suggests that YUV444 should be used if high quality is desired.

### A.7    INFLUENCE OF DATA SHUFFLING

In Section 1, we highlight that GSCodec Studio directly uses the I-frame PLAS index for the P-frame image, exploiting the assumption that the arrangement of the I- and P-frame primitives demonstrates explicit spatial correspondence. The MPEG tracked GS data provides this spatial correspondence naturally due to its generation process (Jeong et al., 2025); therefore, GSCodec Studio reports impressive compression results on the vanilla MPEG tracked dataset. We test the GSCodec Studio with and without data shuffling on the MPEG tracked dataset. The results are shown in Fig. 13. We see that the GSCodec Studio reports obviously poorer performance when data shuffling is applied.

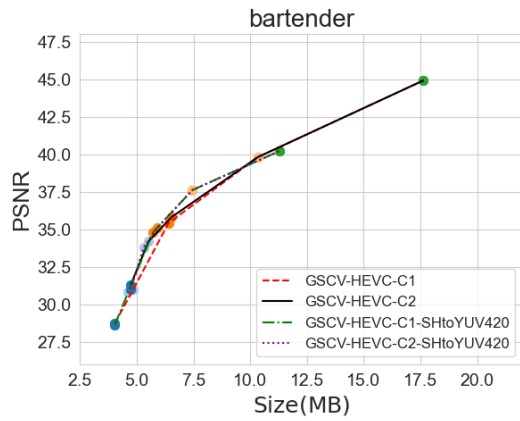

Figure 12: Additional RD curves of GSCV on tracked "bartender".

Specifically, increasing the bitrate by four times is required for achieving the same PSNR compared with the no-shuffle case. In order to ensure I- and P-frame alignment, GSCodec Studio uses primitive padding rather than pruning for the square 2D grid requirement of PLAS, considering that even a minor pruning difference of the I- and P-frames will result in incorrect overall index alignment and the invalidation of index inheritance. The results of GSCodec Studio with pruning are also illustrated in Fig. 13. The obtained observation is close to the shuffle case. Based on the finding from Light-Gaussian (Fan et al., 2024) that GS might have many redundant primitives due to the incomplete densification mechanism, the downstream codec should be compatible with primitive pruning for more flexible rate control, which also highlights the significance and value of the proposed GSCV. Note that if tracked information is available for GSCV (i.e., disable Inter-PLAS), GSCV can still report better performance than GSCodec Studio because GSCV has better compression efficiency with the same PLAS image input as shown in Section A.9.

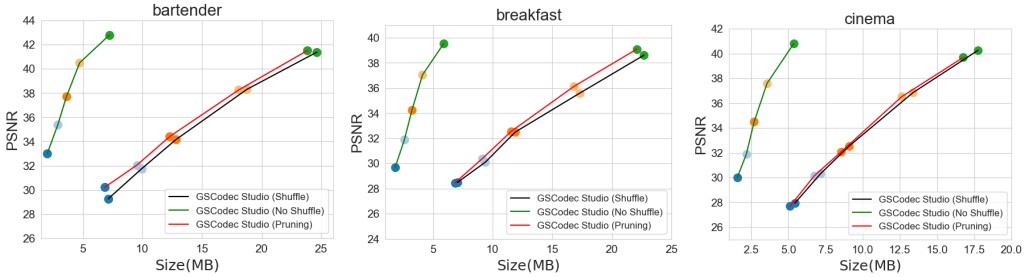

Figure 13: Influence of data shuffle on GSCodec Studio on tracked dataset.

## A.8 EFFECTIVENESS OF ANCHOR-BASED PLAS REFINEMENT

As the prerequisite for realizing efficient inter-frame PLAS image prediction, anchor-based PLAS refinement is important for aligning I- and P-frames. However, different from the vanilla PLAS, which adopts a progressive coarse-to-fine blue image generation strategy to generate smooth intra-frame images, using an I-frame PLAS image as a target to refine the P-frame is more challenging and results in extended computation time requirements. For example, the vanilla PLAS on "bartender", which has around half a million primitives, requires around 30s GPU time, while anchor-based PLAS refinement requires around 70s GPU time. Inspired by the PLAS loss and block size curves shown in Fig. 4, we find that adapting the block size decay factor $\tau$ can result in an improved balance between P-frame image compression effectiveness and generation speed. The default $\tau$ is 0.95 in the vanilla PLAS. We try $\tau = 0.8, 0.7, 0.6$ and report the RD curve and PLAS time in Fig. 14. When reducing $\tau$ from 0.95 to 0.6, we can save around half the GPU time with a minor performance decrease. Therefore, we suggest choosing a proper $\tau$ for P-frame image generation if computation

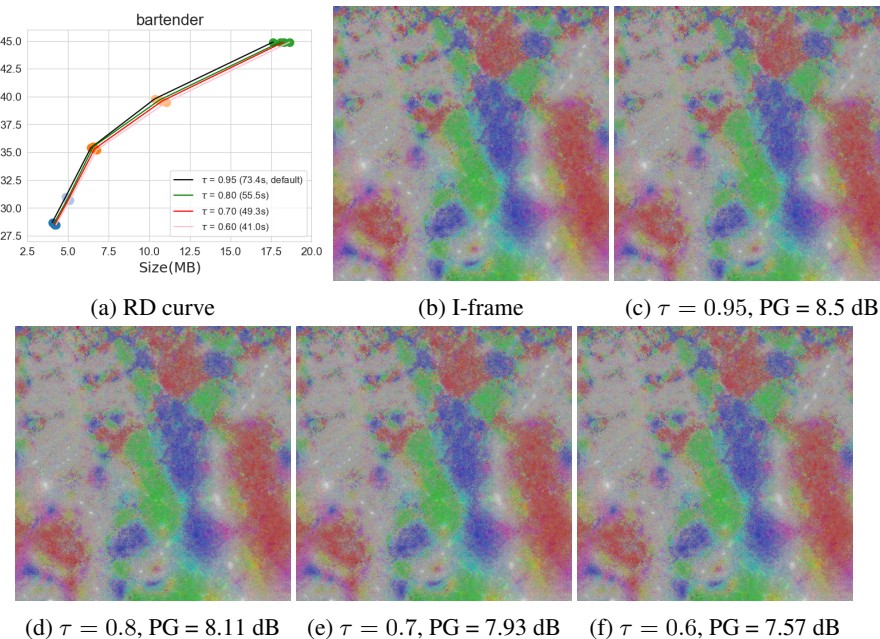

(a) RD curve      (b) I-frame      (c) $\tau = 0.95$, PG = 8.5 dB

(d) $\tau = 0.8$, PG = 8.11 dB     (e) $\tau = 0.7$, PG = 7.93 dB     (f) $\tau = 0.6$, PG = 7.57 dB

Figure 14: Ablation study of anchor-based PLAS refinement.

complexity is emphasized in certain use cases. Besides, we also explore the influence of using a dynamic $\tau$ according to the GS sequence length in Section A.12.

## A.9 ALL-INTRA PERFORMANCE

The proposed GSCV is designed specifically for the GS sequence, while remaining compatible with single-frame compression. We use all-intra mode for GSCV to test the first frame of MPEG "bartender", and compare it with GSCodec Studio (all-intra mode), GPCC v1, HGSC (Huang et al., 2025), and FCGS (Chen et al., 2025a). We find that: 1) The FCGS reports obviously better performance than other methods. The reason is that FCGS is a learning-based method with complex context models. It uses a mask to divide the primitives into different groups and compress them in either the explicit or latent domain. Besides, the FCGS requires GPUs to realize data compression, which is expensive as reported in (Tian et al., 2025). 2) HGSC also requires GPUs to do the primitive pruning based on rendering results as one step of data compression. 3) GSCV, GSCodec Studio, and GPCC v1 are based on CPU, which is lightweight. Although GSCV and GSCodec require PLAS that is based on GPUs, this process is generally not regarded as part of the compression, particularly taking into account that PLAS images may be produced in the course of GS generation or directly thereafter (Morgenstern et al., 2024). 4) GSCV reports better all-intra performance than GSCodec Studio, indicating that GSCV can still offer better inter-prediction results if using the same PLAS videos as input.

## A.10 INFLUENCE OF PLAS FEATURE CHANNEL

In Section 6, we use all the 59 GS features for Inter-PLAS image generation. The time of Inter-PLAS and compression ratio are highly related to the feature channels used in PLAS. To highlight this problem, three cases are proposed to generate PLAS images: 1) using all the features; 2) using coordinates and color DC, i.e., six channels; and 3) using all the attributes except color SH, i.e., fourteen channels. The results are shown in Fig. 16.

We see that employing more features leads to improved compression efficiency. However, more features result in more GPU time for Inter-PLAS. Using "bartender" as an example, applying Inter-PLAS to the I- and P-frames in the three cases considered takes 30.5s/73.4s, 7.9s/22.6s, 12.6s/32.0s. Therefore, besides using $\tau$ to control the Inter-PLAS complexity as reported in Section A.8, controlling the feature channels used in PLAS also deserves more investigations.

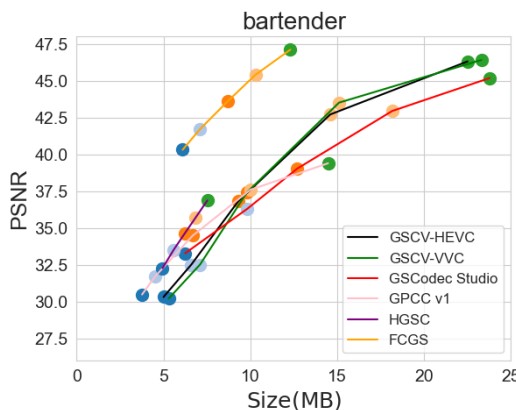

Figure 15: RD curves of single-frame results.

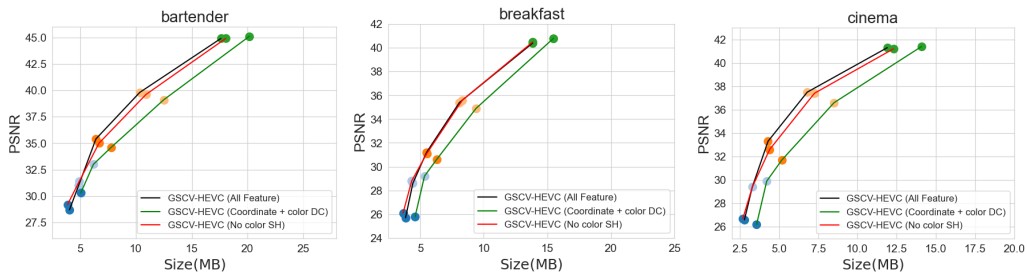

Figure 16: Influence of feature channels on Inter-PLAS

## A.11 INFLUENCE OF COMPRESSION SEQUENCING

In Section 6, we choose the default "encoder_randomaccess" configuration to compress a GoP of PLAS images, in which the first frame is regarded as the I-frame, all the other frames are B-frames (i.e., bidirectionally predicted-coded frame) with compression sequencing as 0-4-2-1-3-6-5-7. We explore the influence of coding sequencing on final results in this section. We choose HEVC "encoder_low_delay" configuration and set frames 1 to 7 as P-frames (predictive-coded frame), the coding sequencing is 0-1-2-3-4-5-6-7. The results are shown in Fig. 17. We see that "RandomAccess" coding mode is slightly better than "low_delay" mode with 3.63% RD gains, indicating that using B-frames in compression can improve overall performance. However, introducing P-frames coding, which does not depend on frames in the future moment, generally requires less buffer time, therefore is more friendly for real-time applications.

## A.12 THE INFLUENCE OF DYNAMIC DECAY FACTOR

We have discussed the influence of $\tau$ in Section A.8, where we find larger $\tau$ represents using more iterations for P-frame PLAS image generation, which generally produces a closer image compared with I-frame images, as well as longer computation time. Theoretically, $\tau$ should be increased with the increase of the length of IPPP sequences. The reason is that with the increase of the sequence length, the difference between the P-frames and I-frames increases, which means the Inter-PLAS requires more iterations to refine P-frames. We perform a new ablation study here. For P-frames of "bartender", we set: 1) D1: initial $\tau_0$ as 0.6, and $\tau_t = min(\tau_0 \times 1.1^{t-1}, 0.95), t >= 1$, in which $\tau_t$ gradually increases to 0.95; and 2) D1: initial $\tau_0$ as 0.95, and $\tau_t = max(\tau_0 \times 0.9^{t-1}, 0.6), t >= 1$, in which $\tau_t$ gradually decreases to 0.6. The result is shown in Fig. 18.

We can see that using a fixed $\tau$ of 0.95 reports the best performance, while the case with an increased $\tau$ introduces around 2.12% gain compared with the case that uses a decreased $\tau$. This corroborates our conclusion that for the GS sequences, the P-frames distant from the I-frames exhibit greater

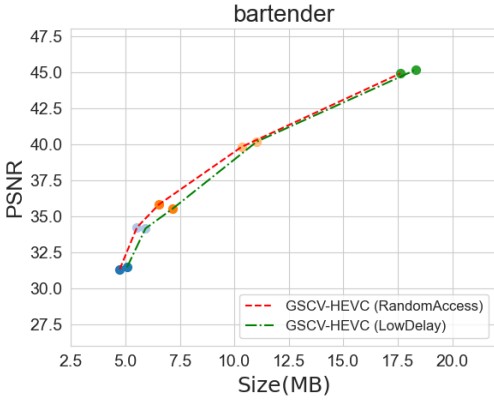

Figure 17: RD curves of different coding sequencing.

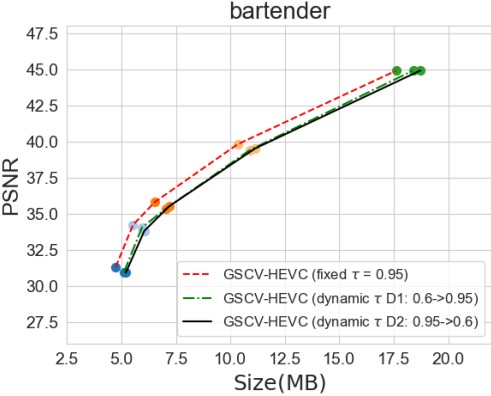

Figure 18: RD curves of dynamic $\tau$.

differences and therefore should be assigned a larger $\tau$. Considering D1 and D2 require a close overall inter-PLAS image generation time, D1 is a better choice given a relatively long GS sequence.

## A.13    RESIDUAL ANALYSIS

Based on the results in Fig. 7, we find the results of HM18.0 and VTM23.11 close to each other, which is in conflict with the results on natural images. Therefore, we illustrate the residual map and the corresponding statistics histogram between frames 0 and 1 of "bartender" color DC and scaling as examples. For a fair comparison, we use the rendered "bartender" with view 9 as the representation of nature images.

Based on the histogram, we further provide fitted Gaussian and Laplace distribution curves. The results are shown in Fig. 19. We see that the natural image reports the smallest standard deviation (Std), and the corresponding histogram reveals smaller entropy. The larger Laplace scale parameter $b$ also reflects that PLAS images demonstrate a more pronounced heavy-tail behavior, indicating more challenging compression conditions. Besides, natural images illustrate obvious motive texture, while PLAS images resemble residual signals that lack semantic content. Consequently, many advanced techniques used in video coding may not function effectively on such images, which can explain the reason that VTM23.11 reports close performance with HM18.0.

## A.14    RD CURVE FOR DIFFERENT COMPONENTS

We illustrate the RD curves of different components in this section. First, we plot the RD curve for color DC, color SH, Opacity, Scaling, and Rotation of method in Fig. 9 as shown in Fig. 20. For certain regions of some features, a few vertically varying points appear because two adjacent bitrate points use the same QP value, while other attributes adopt different QPs.

To better visualize the influence of different components on quality, we perform new experiments: give a certain component, we fix the QP of all the other components as 0, and set five different QP for this certain component. The QP selection is shown in Table 3, and the corresponding RD curves are shown in Fig. 21.

We see that rendered quality is more sensitive to opacity, scaling, and rotation, considering smaller QPs of these components report close PSNR with larger QPs of color DC and SH. Therefore, we choose relatively smaller QP for opacity, scaling, and rotation in our main experiment results, i.e., Fig. 7.

## A.15    BIT ALLOCATION OPTIMIZATION FOR COORDINATE

Based on Fig. 10, we find that coordinates occupy a relatively large bitrate due to the lossless compression, specifically for the LP part. Therefore, we further study this problem and give a new

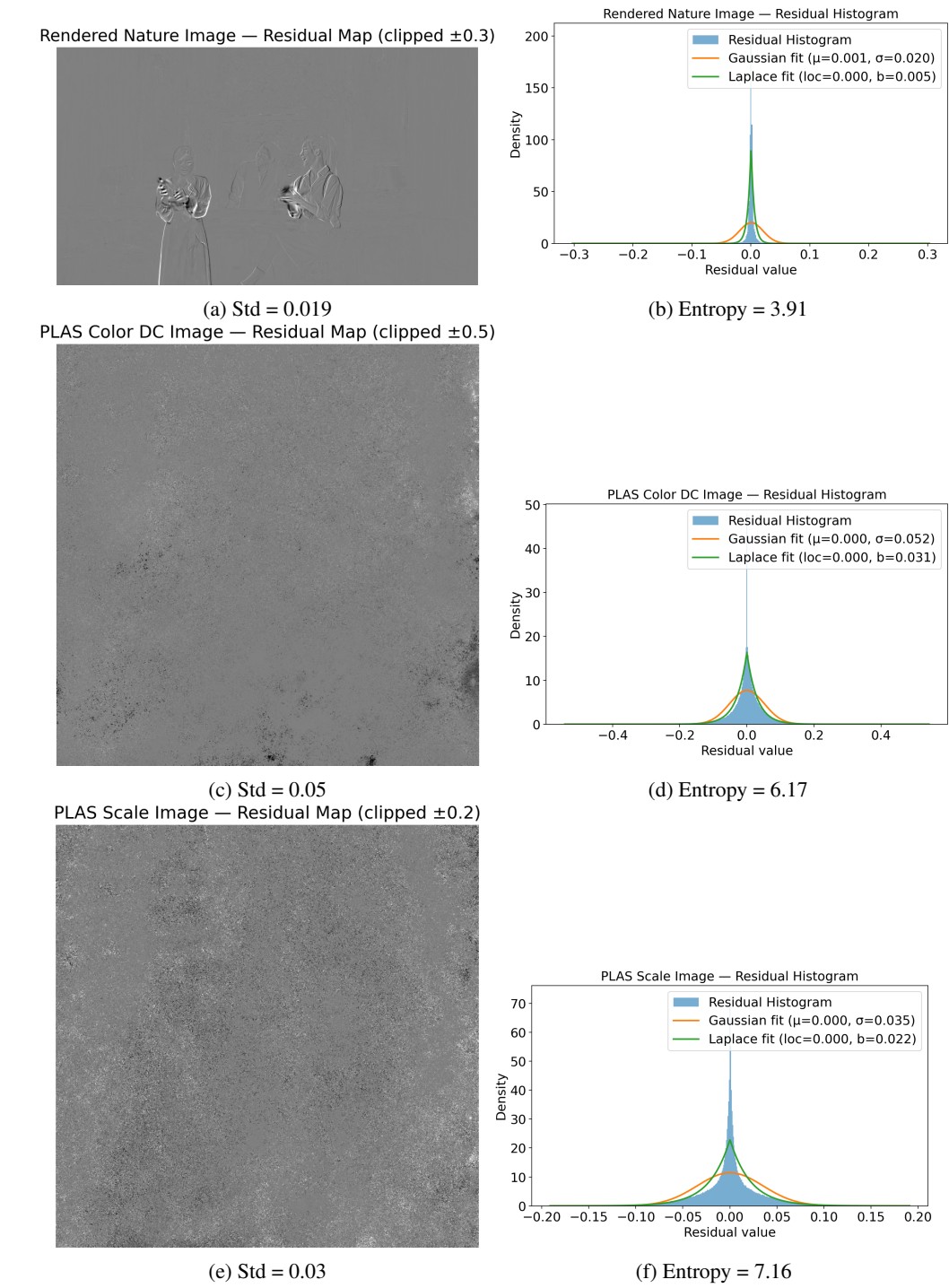

Figure 19: Illustration of residual map and statistic histogram.

solution for coordinate compression. After visualization, we find the LP part of the coordinates is close to a noise map without any texture. Therefore, using a video codec to compress this map has limited performance gain. Therefore, we first try different BD split ratios. For example, given 20 BD coordinates, we choose 10+10, 12+8, and 16+4 for HP and LP parts. Second, we are still using HEVC to compress the HP part, while using LZMA to compress the LP part. Specifically, for inter-prediction with LZMA, we use the I-frame LP map as a reference to calculate the residual for

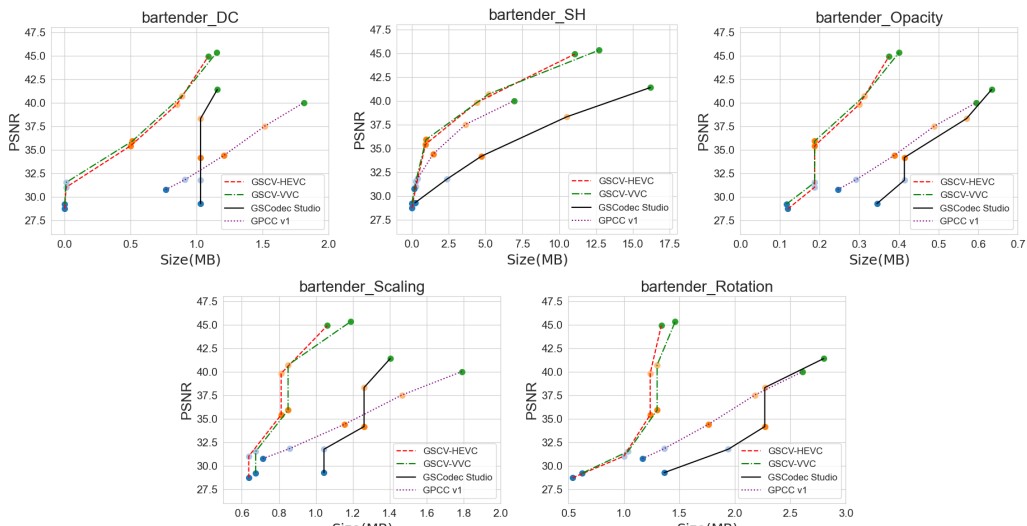

Figure 20: RD curves of different components.

| Component | | QP | | | | |
|---|---|---|---|---|---|---|
| Color DC | | 0 | 22 | 27 | 37 | 47 |
| Color SH | degree1 | 0 | 7 | 17 | 22 | 32 |
| | degree2 | 0 | 12 | 22 | 27 | 37 |
| | degree3 | 0 | 17 | 27 | 32 | 42 |
| Opacity | | 0 | 17 | 22 | 27 | 32 |
| Scaling | | 0 | 7 | 12 | 17 | 22 |
| Rotation | | 0 | 7 | 12 | 17 | 22 |

Table 3: QP of different components

P-frames, followed by LZMA compression. We report the bitstream (Bpp), encoding time, as well as the decoding time results in both all-intra and inter-prediction modes in Table 4.

| Bit Allocation | Mode | HM | HM | | | LZMA | | |
|---|---|---|---|---|---|---|---|---|
| | | HP | LP | All | Time | LP | All | Time |
| 10:10 | All-intra | 13.80 | 31.50 | 45.30 | 3.52/0.26 | 32.00 | 45.80 | 0.92/0.08 |
| | Inter | 10.10 | 30.70 | 40.80 | 16.53/0.18 | 31.96 | 42.06 | 3.64/0.069 |
| 12:08 | All-intra | 19.50 | 25.60 | 45.10 | 3.44/0.22 | 24.00 | 43.50 | 0.36/0.005 |
| | Inter | 15.70 | 25.50 | 41.20 | 16.95/0.15 | 25.00 | 40.70 | 1.59/0.046 |
| 16:04 | All-intra | 31.80 | 13.60 | 45.40 | 3.50/0.18 | 12.50 | 44.30 | 0.44/0.04 |
| | Inter | 28.80 | 13.60 | 42.40 | 18.61/0.11 | 14.30 | 43.10 | 0.9/0.048 |

Table 4: Bit allocation optimization of the coordinate. All = HP + LP

We see that using 12:8 BD split is better than 10:10 and 16:4 under all the testing conditions. With a 12:8 split, using LZMA to compress LP map reports better performance than using HM18.0, as well as significantly improving encoding and decoding speed.

A.16    INFLUENCE OF VIDEO CODEC TYPES

In Section 6, we use HM18.0 and VTM23.11 as representations to test the proposed GSCV. Considering both HM18.0 and VTM23.11 are reference software for academic study, we further use FFMPEG libx265 and libx264 as codecs to test the performance of the proposed GSCV. The RD curves are shown in Fig. 22.

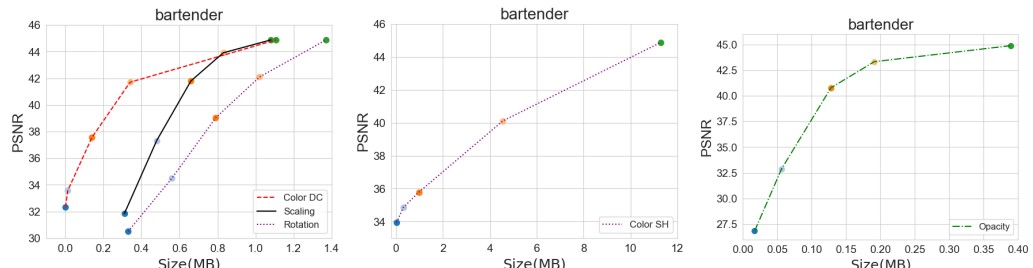

Figure 21: RD curves of different components with smooth QP variation.

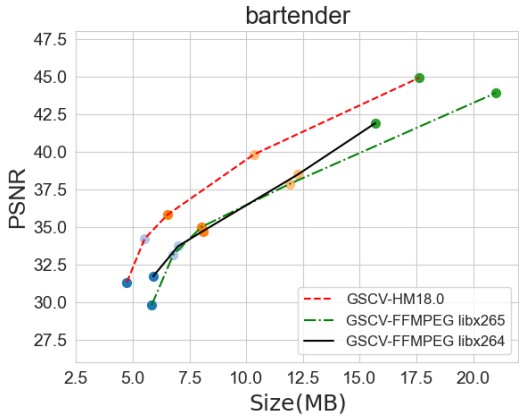

Figure 22: RD curves of using FFMPEG codecs

We see that: 1) the performance of FFMPEG libx265 and libx264 is close. Considering that the same phenomenon also occurs in HM18.0 and VTM23.11, in conjunction with our residual analysis in Section A.13, we think the main reason is that the characteristics of PLAS image are different from natural images, resulting in some advanced compression modules being ineffective for PLAS image compression. 2) HM18.0 reports obviously better performance than FFMPEG libx265, while they both follow the same HEVC (H.265) standard: HM serves as the official reference model for algorithm verification, while libx265 is an optimized engineering realization designed for practical and efficient video compression. This inspires us to give the following conclusion: although some compression modules are improved from AVC to VVC, not all the improvements are useful for PLAS images. There are two analyses corresponding to these results:

• First, the mode of intra prediction increases from 9 (AVC), 35 (HEVC), to 69 (VVC). However, considering the PLAS image does not have semantic texture due to the pixel-level permutation, the increase of intra prediction mode might not improve the compression ratio significantly.

• FFMPEG libx265 uses early skip/ termination for coding unit (CU) partition. Considering PLAS is a progressive method that generates local smoothness images, and the smoothness range and degree are not even for the whole PLAS maps, trying more CU partitions would help the codecs choose the best RD cost, which might be the main reason that libx265 reports obviously worse performance than HM18.0

Although FFMPEG codecs report inferior performance to HM/VTM, they demonstrate obviously higher coding and decoding speed, as shown in Table 5. We see that 1) video-based codecs report faster decoding speed than point cloud based codecs; 2) with the same codec type as libx265, the proposed GSCV reports faster encoding and decoding speed than GSCodec Studio, indicating the advantage of the proposed PLAS image generation method, as well as the framework of the whole pipeline. Therefore, for low delay use cases, we suggest using FFMPEG codecs on the GSCV framework.

| Codec | GSCV-HM18.0 | | GSCV-VTM23.11 | | GSCV-libx265 | | GSCV-libx264 | | GSCodec Studio | | GPCC v1 | |
|---|---|---|---|---|---|---|---|---|---|---|---|---|
| Rate Point | enc | dec | enc | dec | enc | dec | enc | dec | enc | dec | enc | dec |
| 5 | 1183.5 | 3.9 | 11288.9 | 4.2 | 4.6 | 1.2 | 0.8 | 0.6 | 8.9 | 1.4 | 304.1 | 86.0 |
| 4 | 1012.5 | 2.9 | 11234.2 | 3.3 | 4.1 | 0.9 | 0.7 | 0.6 | 8.5 | 1.4 | 298.0 | 88.4 |
| 3 | 540.3 | 2.1 | 6385.6 | 2.3 | 2.8 | 0.7 | 0.6 | 0.5 | 7.3 | 1.3 | 291.7 | 86.7 |
| 2 | 353.4 | 1.6 | 3482.8 | 1.7 | 2.3 | 0.6 | 0.5 | 0.5 | 6.4 | 1.3 | 291.9 | 86.0 |
| 1 | 301.5 | 1.3 | 2391.8 | 1.2 | 1.9 | 0.5 | 0.5 | 0.5 | 5.2 | 1.1 | 293.2 | 85.4 |

Table 5: Encoding and decoding time of different codecs on one GoP (8 frames)

### A.17 COMPARISON WITH A-3DGS BASED METHOD

Different from the I-3DGS track, the A-3DGS track uses the multiview images as input and generates compact 3D GS data via long-time optimization on GPUs. Generally, A-3DGS presents a higher compression ratio than I-3DGS due to the end-to-end training with RD loss. In this section, we compare two I-3DGS methods, i.e., GSCV-HEVC and GSCodec Studio, with four A-3DGS methods: HAC (Chen et al., 2024a) and CompGS (Liu et al., 2024) for the all-intra method, and 4DGS (Wu et al., 2024) and 4DGC (Hu et al., 2025) for the inter-based method. For a fair comparison, we use the multiview images as ground truth to test the quality score of GSCV-HEVC and GSCodec Studio, the results are shown in Fig. 23.

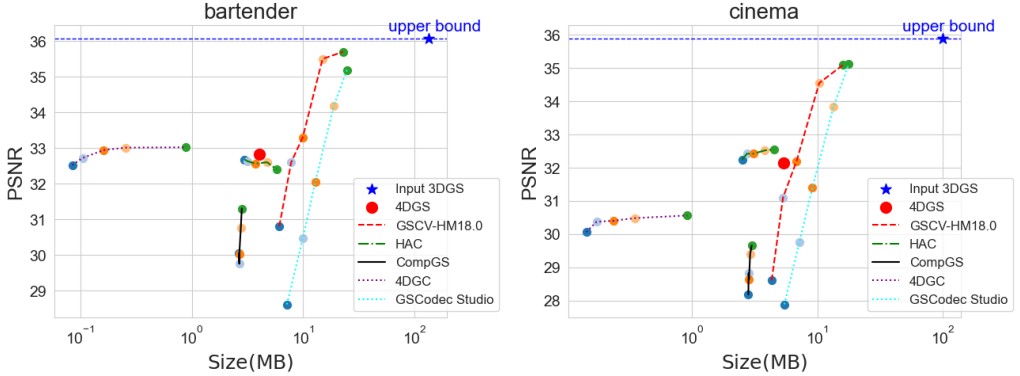

Figure 23: RD curves of A-3DGS methods.

We see that 4DGC reports the best compression ratio, followed by HAC, CompGS, 4DGS, and two I-3DGS methods. However, we find A-3DGS methods have two weaknesses based on the RD curve: 1) regarding the quality of the input 3D GS samples as an upper bound, A-3DGS methods generally demonstrate lower quality and smaller bitrate range; 2) some A-3DGS methods fails to consistently produce a monotonic curve in which the quality increases with the bitrate, due to each rate point needs to be trained from scratch and the optimization process involves a certain degree of randomness.

### A.18 INFLUENCE OF USING SANDWICHED NETWORK

Considering the characteristics of PLAS images are obviously different from natural images, which can limit the performance of canonical video codecs, we try a "Sandwich network" to improve the video codec efficiency (Guleryuz et al., 2024). Specifically, the Sandwich network consists of three parts: the preprocessing network (PreN), the JPEG proxy, and the postprocessing network (PostN). An end-to-end training is conducted based on an established dataset. The RD loss is used to train the PreN and PostN, in which the bitrate is approximated by the JPEG proxy. The trained PreN can convert the target image into a bottleneck image whose data distribution is more friendly for canonical image/video codecs. The practical compression and decompression are performed on the bottleneck image, and the PostN can recover the bottleneck image into the original version.

Three prevalent datasets, i.e., Mip-NeRF360 (Barron et al., 2022), deep blending (Hedman et al., 2018), and tanks&temples (Knapitsch et al., 2017) were applied to generate 3D GS samples and corresponding PLAS images, and build the training dataset. Considering that PLAS images for

different components exhibit different characteristics, and each type of GS attributes needs a separate Sandwich network, in this part, we only used color DC PLAS images as a representation. For PreN and PostN, we used a lightweight UNet consisting of a 4-block encoder and a 5-block decoder. Each encoder block contains two 3×3 Conv-ReLU layers followed by simple pixel-subsampling that halves the spatial resolution, with increasing channel sizes (32→64→128→256). The decoder mirrors the encoder structure, using two 3×3 Conv-ReLU layers per block, bilinear upsampling by a factor of two, and skip-connections that concatenate encoder features at matching scales. The decoder uses filter sizes 512→256→128→64→32. A final 3×3 convolution layer projects the result to the desired number of output channels.

For testing, we use "cinema" as a representation; besides color DC, we keep other attributes lossless. Considering that the Sandwich network using JPEG proxy to train the PreN and PostN, we adopt JPEG (Wallace, 1991) as the codec in the proposed GSCV to better reflect the effectiveness of the Sandwich network. The RD curves obtained were given in Fig. 24. We observed that: 1) the use of FFMPEG libx265 reports better performance than JPEG in the low bitrate region, while JPEG is superior in the high bitrate range; 2) using the Sandwich network followed by the JPEG offers the best performance for the whole bitrate range, indicating the effectiveness of the Sandwich network. Future work is to design Sandwich networks for different GS attributes.

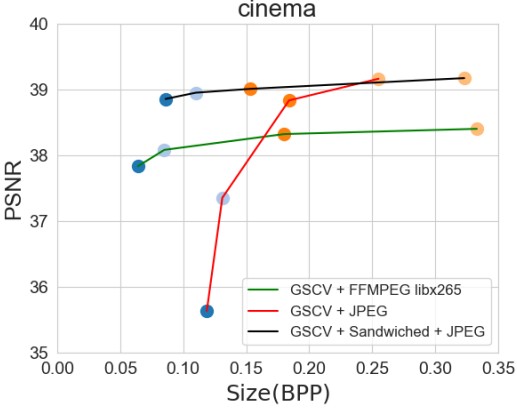

Figure 24: RD curves of using the Sandwich network.

### A.19 RESULTS ON MORE SEQUENCES

In this section, we construct two additional GS sequences for testing our method. Specifically, we use "dance_dunhuang_pair" (dance) from PKU-DyMVHumans (Zheng et al., 2024) and "basketball" from AVS-VRU (AVS, 2024), as shown in Fig. 25.

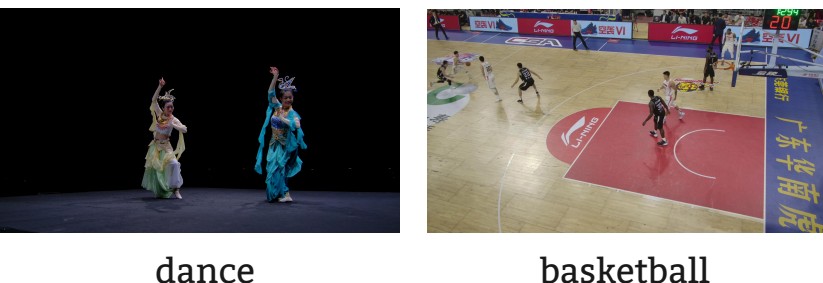

dance                    basketball

Figure 25: Snapshot of "dance" and "basketball"

These two sequences only provide multiview videos; therefore, we use the first 8 frames from the video to generate colmap file, as well as 3D GS samples. We follow the instructions of the MPEG document (Jeong et al., 2025) to merge the per-frame sparse point clouds into a unified one, followed

by a uniform downsampling as the input for 3D GS generation. Then, we train a canonical 3D GS model with ground truth images from all viewpoints and time points. Finally, for each time point, we use the canonical 3D GS model as input to perform attribute fine-tuning with ground truth images.

We test the proposed GSCV-HEVC, GSCodec Studio, and GPCC v1 on the generated ten sequences, the RD curve are shown in Fig. 26. We see that the proposed GSCV demonstrates obvious better performance than GSCodec Studio and GPCC v1, which reveals the superiority of the Inter-PLAS, as well as the compression pipeline.

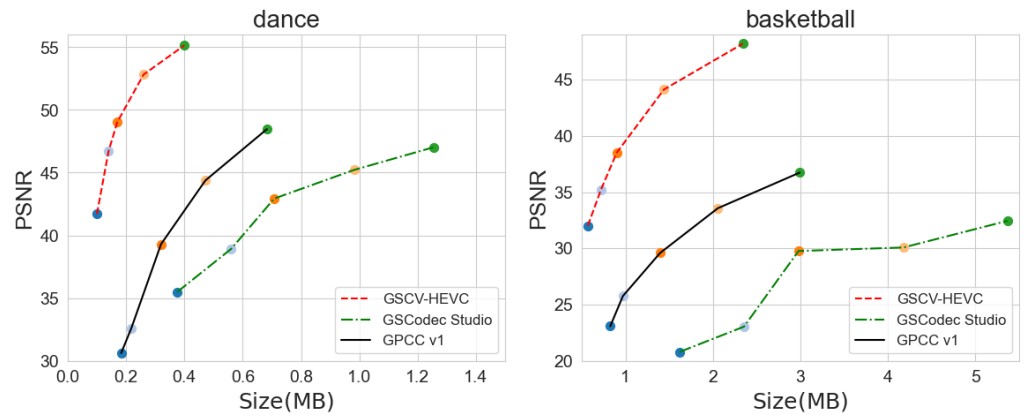

Figure 26: RD curves on "dance" and "basketball".

## A.20 SAMPLE VISUALIZATION

We give the ground truth, rates 5, 3, and 1 of three MPEG tracked GS sequences in Figs. 27 to 29.

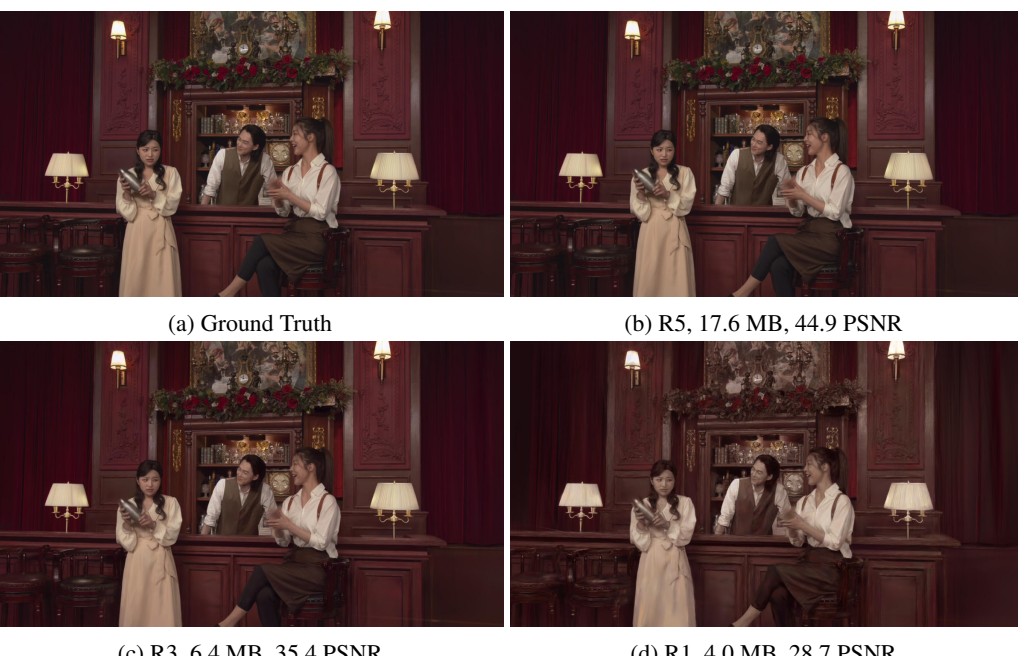

(a) Ground Truth      (b) R5, 17.6 MB, 44.9 PSNR

(c) R3, 6.4 MB, 35.4 PSNR      (d) R1, 4.0 MB, 28.7 PSNR

Figure 27: Snapshot of MPEG "bartender".

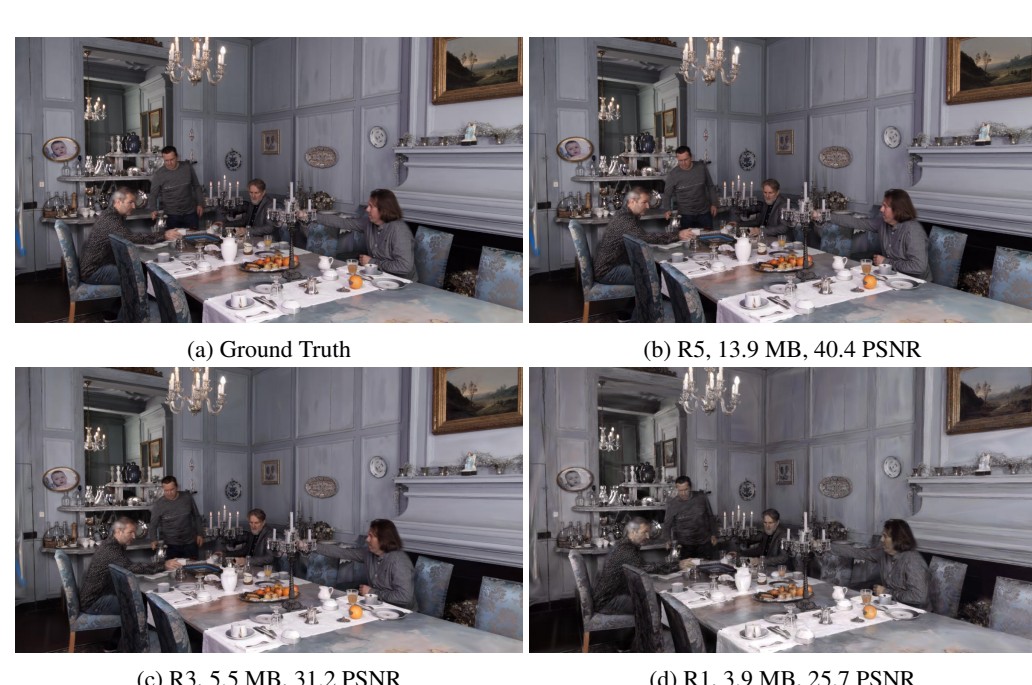

(a) Ground Truth         (b) R5, 13.9 MB, 40.4 PSNR

(c) R3, 5.5 MB, 31.2 PSNR         (d) R1, 3.9 MB, 25.7 PSNR

Figure 28: Snapshot of MPEG "breakfast".

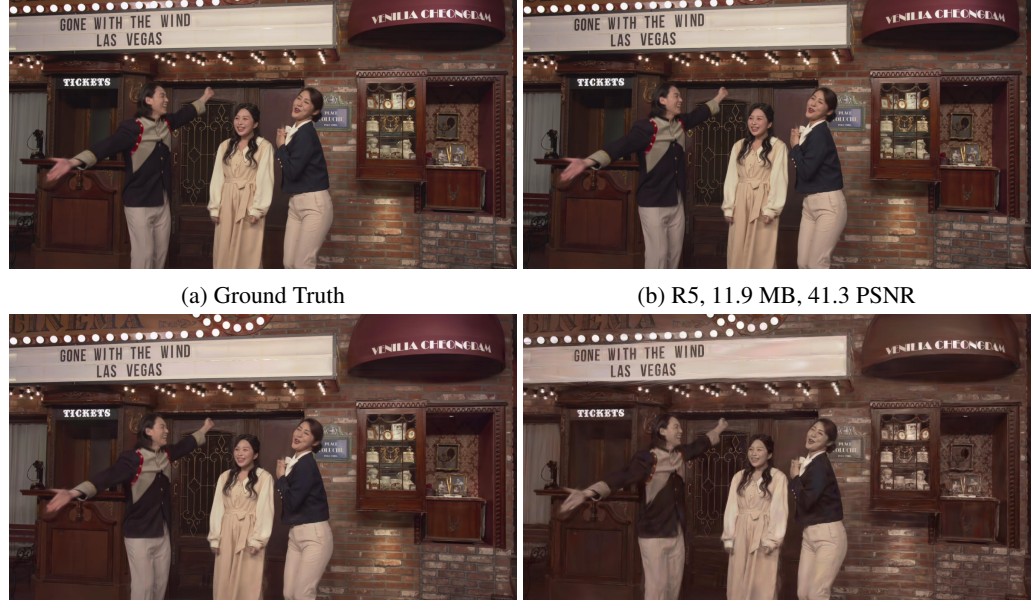

(a) Ground Truth         (b) R5, 11.9 MB, 41.3 PSNR

(c) R3, 4.3 MB, 33.3 PSNR         (d) R1, 2.8 MB, 26.6 PSNR

Figure 29: Snapshot of MPEG "cinema".

