# OpenReview forum: "GSCV: Compressing Gaussian Splatting Sequence with Video Codec"
_ICLR.cc/2026/Conference — Submitted to ICLR 2026_

### Official Review · Reviewer_An3v · 2025-10-26

**Soundness:** 3
**Presentation:** 3
**Contribution:** 3
**Rating:** 6
**Confidence:** 4

**Summary:**

## Summary
* This paper proposes a more effective approach towards GS compression using video codec.
* The proposed method consists of an improved inter-frame consistency module and a support for higher bit depth, which enhance the efficiency and upperbound of video codec.
* The authors provide convincing empirical evidence to support their claim.

**Strengths:**

## Strength
* The authors have shown that their initialization and refinement successfully stablize the GS representation between I and P frames, through clear demonstration and examples.
* The empirical result is convincing as the advantage of the proposed method over baseline is very obvious as shown in Fig 7. Besides, the authors have show that their approach outperforms other PC codec such as GPCC v1 in high bitrate regime.

**Weaknesses:**

## Weakness
* The adoptation of heavy video codec such as HM and VTM might increase the encoding time in a non trivial way. The adoptation of YUV444 format and 10 bits depth might hinders the adoptation of faster variant such as x265. Then, it is better to discuss the temporal complexity of different methods in Fig 7 for fair comparsion.
* It is a little bit surprising to see that VVC is outperformed by HEVC, considering the performance gap on RGB videos. Additional explaination might strengthen the result.
* As the input images are from different domains, it is probabily benefical to consider pre/post processing those images before/after the video codec, such as [One-click upgrade from 2D to 3D: Sandwiched RGB-D video compression for stereoscopic teleconferencing]. Afterall, HM and VTM are tuned for RGB videos.

**Questions:**

## Questions
* The weird relative performance between VTM and HM might comes from the fact that the images domain deviants too much from RGB. Is that possible that simpler codec, such as H264 works as well?

---

> ### Author Response · Authors · 2025-11-19
> **Response for weakness**
>
> # For Weakness 1:
>
> Thanks for your valuable comments. We would like to point out that **YUV444 and 10 bits are readily supported by x265**. To illustrate this, we used FFMPEG libx265 and libx264 in our framework and obtained new RD curves for your reference in Appendix A.16. For a fair comparison of the temporal complexity of different methods, we listed the average encoding and decoding time of using VTM, HM18.0, libx265, libx264 in the proposed pipeline, as well as GSCodec Studio and GPCC v1, in Table 5 of Appendix A.16. Based on the newly attained RD curves and temporal complexity of different codecs, we provided more analysis about why HM18.0 reported close performance as VTM23.11, and why the proposed Inter-PLAS and framework of GSCV have superiority over other video-based anchors. Please refer to Appendix A.16 in our revised paper for details. In summary, our pipeline is compatible with all the SOTA video codecs, which is one of our main contributions. We can choose different codecs to accommodate the requirements of downstream tasks, e.g., a higher compression ratio or fast coding speed.
>
> # For Weakness 2:
>
> Thanks for your valuable comment. To better understand the observation, we add a new section, Appendix A.13, to analyze the residual of PLAS images. The residual map and histograms revealed that **PLAS images exhibit a more pronounced heavy-tail behavior, and the residuals do have semantic texture as natural images**. Therefore, we concluded that the PLAS images are different from natural images, rendering some advanced compression modules ineffective for PLAS images. Please check section A.13 for details. The results of using FFMPEG libx265 and libx264 in Appendix A.16, which are close to each other, can also support this finding.
>
> # For Weakness 3:
>
> We agree with the reviewer and **also pointed out this aspect in the Conclusion section of the original paper**. Furthermore, we added a new experiment in Appendix A.18. The Sandwich network consists of three parts: the preprocessing network (PreN), the JPEG proxy, and the postprocessing network (PostN). An end-to-end training is conducted based on an established dataset. The RD loss is used to train the PreN and PostN, in which the bitrate is approximated by the JPEG proxy. Three prevalent datasets, i.e., Mip-NeRF360, deep blending, and tanks\&temples were applied to generate 3D GS samples and corresponding PLAS images, and build the training dataset. Considering that PLAS images for different components exhibit different characteristics, and each type of GS attributes needs a separate Sandwich network, in this part, we only used color DC PLAS images as a representation. For PreN and PostN, we used a lightweight UNet consisting of a 4-block encoder and a 5-block decoder. Each encoder block contains two 3×3 Conv-ReLU layers followed by simple pixel-subsampling that halves the spatial resolution, with increasing channel sizes (32→64→128→256). The decoder mirrors the encoder structure, using two 3×3 Conv-ReLU layers per block, bilinear upsampling by a factor of two, and skip-connections that concatenate encoder features at matching scales. The decoder uses filter sizes 512→256→128→64→32. A final 3×3 convolution layer projects the result to the desired number of output channels.
>
> For testing, we use “cinema” as the representation; besides color DC, we keep other attributes lossless. Considering that the Sandwich network using JPEG proxy to train the PreN and PostN, we adopt JPEG as the codec in the proposed GSCV to better reflect the effectiveness of the Sandwich network. The RD curves obtained were given in Fig. 24. We observed that: 1) the use of FFMPEG libx265 reports better performance than JPEG in the low bitrate region, while JPEG is superior in the high bitrate range; 2) using the Sandwich network followed by the JPEG offers the best performance for the whole bitrate range, indicating the effectiveness of the Sandwich network. Future work is to design Sandwich networks for different GS attributes.

---

> ### Author Response · Authors · 2025-11-19
> **Response for question**
>
> # For Question:
>
> Thanks for your comment. In our response to previous questions, we used the residual analysis to show that PLAS images are quite different from natural images (see Appendix A.13). Besides, we tested more codecs such as FFMPEG libx265 and libx264 for compressing our Inter-PLAS images under the proposed GSCV, which is easy to implement considering our pipeline is compatible with all the SOTA video codecs. The results are shown in Appendix A.16. It was found that the performance of FFMPEG libx265 and libx264 is similar, while HM18.0 offers obviously better performance than libx265. Considering that libx265 and HM18.0 are under the same standard (HEVC), and libx265 is a lightweight version of HM via using Early Skip on CU partition and other simplified tools, **we can conclude that not all the modules can improve PLAS compression ratio**. We gave more analysis in Appendix A.16, as well as a coding time comparison between codecs. Please check our revised paper for more details.

---

> ### Author Response · Authors · 2025-11-27
> **Looking forward for your valuable feedback**
>
> Dear reviewer **An3v**,
>
> Thank you for your time to review our manuscript and valuable feedback and recognition. We have carefully addressed all the comments and concerns raised, as reported in our detailed response and revised manuscript.
>
> We are looking forward to your second round of review.
>
> Best,
>
> Authors

---

### Official Review · Reviewer_qDJS · 2025-10-29

**Soundness:** 2
**Presentation:** 3
**Contribution:** 2
**Rating:** 4
**Confidence:** 4

**Summary:**

The paper introduces GSCV, a novel method for compressing 3D Gaussian Splatting sequences using standard video codecs. Unlike previous approaches that rely on optimization-based compression (A-3DGS), GSCV focuses on compressing the trained GS data (I-3DGS).The core innovation is Inter-PLAS, an improved version of the Parallel Linear Assignment Sorting (PLAS) method and anchor-based refinement, which ensures that consecutive GS frames (I- and P-frames) produce similar images, enhancing inter-frame prediction in video codecs like HEVC and VVC. Experiments on MPEG datasets show that GSCV outperforms existing video- and point-cloud-based methods in both tracked and semi-tracked GS sequences at some bitrate region.

**Strengths:**

1. Effective Inter-Frame Compression.  Spatial-Stable Initialization (SSI) and anchor-based refinement reduce randomness in PLAS. These modules  improve frame similarity, leading to better inter-frame-prediction

2. Compatibility with Standard Codecs. Compression techniques for pre-trained Gaussian Splatting models are practical in certain scenarios, while leveraging existing video encoders also offers better adaptability, making them applicable in environments lacking specialized acceleration hardware.

**Weaknesses:**

1. Limited Compression Efficiency on PLAS Images. PLAS-generated images are structurally different from natural images, limiting the full potential of standard video codecs.

2. The experimental comparisons are insufficient. Although the technical paradigms are not entirely identical, there are already several studies on Gaussian Splatting sequences or 4D Gaussians. Conducting more comparisons on more dataset would more comprehensively reflect the performance of the proposed method in this paper.

some previous work:

[1] 4D Gaussian Splatting for Real-Time Dynamic Scene Rendering

[2] 4DGC: Rate-Aware 4D Gaussian Compression for Efficient Streamable Free-Viewpoint Video

**Questions:**

1. Generally, methods leveraging existing codecs, even if not achieving optimal overall performance, still significantly outperform training-based approaches in certain aspects, such as encoding time. This allows for some compromise in rate-distortion (RD) performance. However, PLAS-based methods still require thousands of iteration steps—does this undermine the practical value of such approaches? Furthermore, could some encoding time results be provided in the experiments?

2. In the low-bitrate region of the rate-distortion curve (bartender, cinema), the performance of the GSCV method falls below that of the GPCC all-intra mode. Does this indicate that the proposed inter-frame model still has certain limitations?

---

> ### Author Response · Authors · 2025-11-19
> **Response for weakness**
>
> # For weakness 1:
>
> Thanks for this insightful comment. We agree with the reviewer that PLAS images are different from natural images, making it hard to fully exploit the potential of standard video codecs. But we still chose to pursue this direction due to 1) these standard video codecs having been widely adopted in current products, and 2) **the inspiration from the MPEG I-3DGS track that also aims at using standard video or point cloud codecs to realize GS compression with PLAS as the video-based anchor**. Our work significantly improved PLAS, especially in the use case of compressing the GS sequences without tracked information. To address your comment, **we did a new experiment to further improve the performance of standard video codecs on PLAS images through training a Sandwich network**. The results are summarized in Appendix A.18 of the revised paper. The Sandwich network can convert the PLAS images to bottleneck images which are easier to code by standard video codecs. The results are summarized in Appendix A.18 of the revised paper. The observation is that using a Sandwich network can enhance compression efficiency, and this topic deserves more attention in future work. In summary, although the current PLAS image is different from natural images and limits the full potential of standard video codecs, our research is meaningful and contributes to solving this problem.
>
> # For weakness 2:
>
> Thanks for the comment. **We would like to clarify that methods based on 4D Gaussians, which belong to the A-3DGS category, are compression techniques fundamentally distinct from our method**. For A-3DGS methods, both the input and ground truth are multi-view images/videos. Our method, on the other hand, takes the generated GS samples as the input, and the ground truth is the rendered images from the input GS samples. Besides, A-3DGS methods require a complex optimization process carried out over GPUs to realize compact GS generation, while our method can be executed on CPUs with generated GS samples which is a less stringent requirement on the hardware. Thus, a direct comparison in terms of the compression ratio only would be unfair.
>
> To address your comment, 1) we evaluated the two all-intra A-3DGS methods, HAC and CompGS, as well as the two 4DGS methods using MPEG colmap data, and re-tested the PSNR of our method and GSCodec Studio using the same ground truth images. The results were reported in Appendix A.17. We found that 4DGC offers the best compression ratio, followed by HAC, CompGS, 4DGS, and the two I-3DGS methods. However, A-3DGS methods exhibit two drawbacks according to the obtained RD curves. 1) With the quality of the vanilla 3D GS samples as an upper bound, A-3DGS methods generally offer lower quality and smaller bitrate range. 2) Some A-3DGS methods fail to consistently yield a monotonic curve (i.e., the rendering quality does not necessarily increase with the bitrate), because each rate point needs to be trained from scratch and the optimization process involves a certain degree of randomness. 2) We constructed two new GS sequences following the instruction of MPEG, i.e., “dance” and “basketball”, and test the proposed GSCV, GSCodec Studio, and GPCC v1 on these two sequences. The results were shown in Appendix A.19. On the new dataset, GSCV still reports the best performance.

---

> ### Author Response · Authors · 2025-11-19
> **Response for question**
>
> # For question 1:
>
> Thanks for the questions. We agree with the reviewer that producing PLAS images does require thousands of iterations. However, in our opinion, **this amount of time should be considered as part of the GS postprocessing time, as GS content cannot be generated in real-time anyway**. PLAS can be implemented on either GPU or CPU, which is also one of the reasons that MPEG adopts this method in the current video-based compression anchor. Besides, we showed the combined encoding and decoding time of HM18.0 in Fig.11 of Appendix A.5. It can be seen that one advantage of using existing codecs is that it is possible to realize real-time decoding with multithreaded computation on CPU, considering that the decoding of each 2D map is independent. To further illustrate this advantage, we used FFMPEG libx265 and libx264 in our pipeline, and reported the RD curves, encoding and decoding time in Appendix A.16 of the revised paper (times in Table are lineraly accumulated of multiple PLAS images, rather than the parallel processing time). We found that 1) video-based codecs report faster decoding speed than point cloud based codecs; 2) with the same codec type as libx265, the proposed GSCV reports faster encoding and decoding speed than GSCodec Studio, indicating the advantage of the proposed PLAS image generation method, as well as the framework of the whole developed pipeline.
>
> # For Question 2:
>
> Thanks for the question. We analyzed this observation in Appendix A.6. We thought that the main reason is the selection of QP for different attributes. Specifically, it was found that both increasing the QP of color DC and using YUV420 for color SH can improve the RD in the low-bitrate region. Thus, it is worthwhile to explore how to realize rate allocation for different GS components corresponding to different bitrate regions. We provided a more detailed RD curves for each component in Appendix A.14. Based on the results in Fig. 21, we noticed that the rendered quality is more sensitive to opacity, scaling, and rotation, considering that smaller QPs for these components lead to close PSNR with larger QPs of color DC and SH. A more comprehensive bit allocation algorithm is needed for optimal GS compression for video codecs, which is one of our future work. We also noted that the residuals between the I- and P-frame PLAS images are different from the natural images, which could be another reason. Based on our analysis in Appendix A.13 and Fig. 19, the residual of PLAS images is noise-like and demonstrates a more pronounced heavy-tail behavior. Therefore, when using a relatively larger QP, the loss becomes larger than natural images with the same QP level. Therefore, another possible future work is on how to further improve the similarity between I- and P-frames without increasing the computational complexity.

---

> ### Author Response · Authors · 2025-11-27
> **Looking forward for your second review**
>
> Dear reviewer **qDJS**,
>
> Thanks for your effort and time to review our paper, your comments have helped us improve our manuscript significantly. We have carefully read your comments and added new experiments to solve your questions. The revised manuscript has been updated accordingly.
>
> We are looking forward to your further comments.
>
> Best,
>
> Authors

---

### Official Review · Reviewer_wnCS · 2025-11-01

**Soundness:** 2
**Presentation:** 2
**Contribution:** 2
**Rating:** 2
**Confidence:** 3

**Summary:**

The paper targets I-3DGS (optimization-free) compression of Gaussian Splatting (GS) sequences using standard video codecs (HEVC/VVC). The core idea is to convert each GS frame to a set of smooth 2D “PLAS images” and then exploit inter-frame prediction in video codecs. The authors identify a weakness in the vanilla PLAS pipeline for sequences, i.e., independent stochastic initializations make I/P frames misaligned, hurting inter prediction. They propose Inter-PLAS, comprising (i) Spatial-Stable Initialization (SSI) via 3D Morton-order sorting plus reuse of the I-frame’s PLAS index as context for the P-frame, and (ii) an anchor-based PLAS refinement that aligns the P-frame’s PLAS image directly toward the I-frame to raise inter-frame similarity. They also introduce a practical bit-depth design: coordinates are quantized to 20-bit and split into two 10-bit images, other attributes are 10-bit, forming 22 attribute videos per GS sequence and enabling compatibility with HEVC/VVC. Experiments on MPEG tracked & semi-tracked GS sequences (bartender, cinema, breakfast) report gains over a video-based anchor (GSCodec Studio) and a point-cloud anchor (GPCC v1).

**Strengths:**

1. The paper squarely addresses the gap in inter-frame GS compression with canonical codecs, aligning with I-3DGS industrial constraints (CPU-centric decoding, codec re-use). It discusses A-3DGS vs I-3DGS and focuses on I-3DGS compression with optimization-free pipelines, together mature video codecs such as H.265/266.

2. Engineering that makes video codecs work. The 10-bit per-attribute plus 20-bit (split) for coordinates is a pragmatic choice that avoids quality ceilings while keeping compatibility (YUV444; careful avoidance of RGB↔YUV rounding). The paper also explains why VVC’s usual advantage over HEVC diminishes on PLAS images, indicating distribution shift vs natural video.

**Weaknesses:**

1. Limited methodological novelty.
The paper presents a straightforward engineering implementation that leverages standard video codecs (HEVC/VVC) for Gaussian Splatting sequence compression. While practical and well-structured, the core contribution—mapping GS attributes into 2D PLAS images and feeding them to an existing codec—is conceptually simple and incremental rather than a fundamentally new compression principle.

2. Incomplete comparison with learning-based and adaptive GS compression.
The work lacks comparisons with A-3DGS methods that optimize Gaussian kernels directly for compression efficiency, as well as with learning-based I-3DGS compressors such as FCGS in the inter-frame setting. Although the appendix briefly touches FCGS in an all-intra setting, obviously sequence-level modification of FCGS produces much better performance.

3. Shallow codec-level modeling.
Although the framework maintains compatibility with HEVC/VVC, it does not perform any explicit rate–distortion modeling or entropy analysis. The system relies entirely on codec defaults without quantifying the optimality of its design choices. The 10-bit attribute and 20-bit coordinate configuration appear heuristic; a principled justification via per-component rate–distortion curves, entropy measurements, or bit-allocation optimization would substantially strengthen the technical rigor.

**Questions:**

Please see in Weaknesses Part.

In addition, Why does VVC ≈ HEVC here (Figure 7)? Have you profiled residual statistics (e.g., heavier tails, non-natural textures) to quantify how PLAS images deviate from natural video?

The layout of statistics in some figures could be improved a lot. It is hard to distinguish some data points there.

---

> ### Author Response · Authors · 2025-11-19
> **Response for weakness 1 and 2**
>
> # For weakness 1:
>
> Thanks for your valuable comments. The proposed solution does appear to be a conceptually simple one. However, its underlying value should not be underestimated. If the existing video codecs, which have been readily supported by major hardware manufacturers, can be combined with effective lightweight techniques to compress GS videos, the corresponding compression pipeline thus has the outstanding potential to be rapidly and widely deployed to speed up the application of 3DGS. **This pathway is seriously evaluated by both industry and standardization organizations such as MPEG. Our work is aligned with this direction**. The proposed SSI and anchor-based PLAS refinement are both lightweight but empirically efficient. Besides, our work also studied how to generate the GS 2D map sequences reliably and best exploit the ability of these codecs. For current video-based GS compression anchors, there are key shortcomings, as we pointed out in the Introduction section of the paper. Addressing these weaknesses deserves considerable attention.
>
> # For weakness 2:
>
> Thanks for the comments. Please allow us to explain. **A-3DGS follows a rather different approach from our work**. Its input is a multi-view image and requires a complex optimization process carried out over GPUs to realize compact GS generation. On the contrary, our method uses the generated 3DGS as the target to be compressed and can be executed on CPU, a less stringent requirement on the hardware, making it more suitable for lightweight devices such as smartphones. This also renders our pipeline more aligned with the MPEG document “m72430 [GSC][JEE6.4-related] On the use case and requirements for lightweight GSC”, which specified GS compression to have “low power, low complexity, and lightweight coding and decoding”. Thus, a direct comparison in terms of the compression ratio only is not fair. Similar arguments hold for learning-based I-3DGS compressors like FCGS, which also need powerful GPUs for deployment.
>
> Besides, we added a new experiment in Appendix A.17 to illustrate the difference between A-3DGS and I-3DGS. We compare two I-3DGS methods, i.e., GSCV-HEVC and GSCodec Studio, with four A-3DGS methods: HAC and CompGS for the all-intra method, and 4DGS and 4DGC for the inter-based method. For a fair comparison, we used the multi-view images as ground truth to test the quality score of GSCV-HEVC and GSCodec Studio. The results were shown in Fig. 23. We see that 4DGC has the best compression ratio, followed by HAC, CompGS, 4DGS, and two I-3DGS methods. However, we find A-3DGS methods have two weaknesses according to the RD curve: 1) regarding the quality of the vanilla 3D GS samples as an upper bound, A-3DGS methods generally demonstrate lower quality and smaller bitrate range; 2) some A-3DGS methods fails to consistently produce a monotonic curve in which the quality increases with the bitrate, due to each rate point needs to be trained from scratch and the optimization process involves a certain degree of randomness.

---

> ### Author Response · Authors · 2025-11-19
> **Response for weakness 3**
>
> # For weakness 3:
>
> Thanks for the comments. We would like to point out that **some discussions on rate-distortion modeling and entropy analysis have been given in Appendices A.6 and A.10**. Specifically, in A.6, we considered the influence of different QP selections on color DC and different YUV modes for color SH. We observed that smaller QPs of color DC and using YUV420 for color SH can contribute to the middle-to-low bitrate range. In A.10, we studied the influence of feature selection of PLAS on the overall performance. We tested three cases: only use xyz+color DC, all features except color SH AC, and all features. We found that the influence of SH AC is limited, while introducing opacity, scaling, and rotation is important. The features which are used in the PLAS process will consequently result in a corresponding smooth 2D map, therefore contributing to the final overall rate-distortion performance. We acknowledged that considering the rich attribute information of GS, the current discussions on the relationships among different attributes are still limited. However, **designing a comprehensive entropy model or bit-allocation optimization is beyond the scope of this paper**. Our scope is two-fold: 1) to establish tracked information for PLAS videos; and 2) to design a video-based pipeline that is compatible with SOTA video codecs with higher quality upper bounds, as well as a better data processing strategy. We plan to further investigate this problem in the future.
>
> 10-bit attributes and 20-bit coordinates are not heuristic choices. There are actually a few MPEG proposals supporting our selection. In “[GSC][JEE6.7-related] a lightweight 3d Gaussian splats coding framework for 1F-VID track with a single video decoder instance. ISO/IEC JTC 1/SC 29/WG 4 m73294, 2025,” 10-bit attributes were suggested. In “[GSC][JEE 6.7] Report on the performance of V-PCC platform for GSC. ISO/IEC JTC 1/SC 29/WG 4 m74012, 2025,” 10-bit attributes were also recommended. Another evidence is from GPCC v1, MPEG WG 7 suggested using 18 bits for coordinates and 12 bits for attributes. This was cited in Page 2 of our paper in lines 74-80 as “MPEG preliminary experiments for GS samples show that the optimal coordinates BD are 14 to 18 with respect to different datasets”. Besides, they thought that “it would be better to use a larger BD during quantization and realize quality-bitstream balance via compression”. Therefore, we chose 20 bits for coordinates and 10 bits for attributes in this study.
>
> We further generated per-component rate distortion curves of the methods tested in Fig.7 and added them to Appendix A.14. For certain regions of some features, a few vertically varying points appear because two adjacent bitrate points use the same QP value, while other attributes adopt different QPs.
>
> To better visualize the influence of different components on the quality, we performed new experiments: given a certain component, we fixed the QP of all the other components at 0, and set five different QP for this certain component and plotted new RD curves in A.14.
>
> We found that the rendered quality is more sensitive to opacity, scaling, and rotation, considering smaller QPs of these components reported close PSNRs with larger QPs of color DC and SH. Therefore, we chose relatively smaller QP for opacity, scaling, and rotation in our main experiment results reported in Fig.7.
>
> Regarding bit allocation optimization, we discussed it in Appendix A.5. It was found that coordinates occupy a relatively large bitrate due to lossless compression, specifically for the LP part. Therefore, we conducted further study and gave a new solution to coordinate compression in Appendix A.15. After visualization, we noted that the LP part of the coordinates is close to a noise map without any texture. Therefore, using a video codec to compress this map has limited performance gain. We then tried different BD split ratios. For example, given 20 BD coordinates, we chose 10+10, 12+8, and 16+4 for HP and LP parts. Second, we were still using HEVC to compress the HP part, while using LZMA to compress the LP part. Specifically, for inter-prediction with LZMA, we used the I-frame LP map as a reference to calculate the residual for P-frames, followed by LZMA compression. We reported the bitstream (Bpp), encoding time, as well as the decoding results in both all-intra and inter-prediction modes in Table 4. It can be seen that using the 12:8 BD split is better than 10:10 and 16:4 under all the testing conditions. With a 12:8 split, using LZMA to compress LP map reports better performance than using HM18.0, as well as significantly improving the encoding and decoding speeds.

---

> ### Author Response · Authors · 2025-11-19
> **Response for question**
>
> # For question 1:
>
> Thanks for your valuable comments. Based on your suggestions, we analyzed the residual statistics of nature images and the PLAS images. We used the rendered image of “bartender” frame 0 and frame 1 under view 9 as the representation of the natural video, and used the PLAS color DC and scaling images as comparison. We illustrated the residual map and corresponding residual statistic histogram, as well as fitted Gaussian and Laplace distribution curves, in Appendix A.13. We see that the natural image reports the smallest standard deviation (std), and the corresponding histogram reveals a smaller entropy. The larger Laplace scale parameter b also reflects that **PLAS images demonstrate a more pronounced heavy-tail behavior, indicating more challenging compression conditions**. Besides, natural images illustrate obvious motive texture, while PLAS images resemble residual signals that lack semantic content. Consequently, many advanced techniques used in video coding may not function effectively on such images, which can explain the reason that VTM23.11 reports close performance with HM18.0.
>
> # For question 2:
>
> We have improved the figures to make them clearer, specifically for Fig. 3.

---

> ### Author Response · Authors · 2025-11-27
> **Looking forward to your further review**
>
> Dear reviewer **wnCS**,
>
> Thank you for taking the time to review our manuscript and for your valuable comments. We have carefully addressed all the questions and conducted new experiments to address your concerns, as reflected in our revised manuscript.
>
> We are looking forward to your further assessment.
>
> Best,
>
> Authors

---

> ### Comment · Reviewer_wnCS · 2025-11-27
> **Feedbacks after rebuttal**
>
> Thanks authors for the detailed responses.
>
> The main contribution of this paper is to realize optimization-free compression of Gaussian Splatting (GS) sequences using standard video codecs. However, as explained in your rebuttal, if VVC and HEVC both may not function effectively on such kind of a data (which results in similar performance of using VVC and HEVC), the proposed framework may not be a convincing one: is using standard video codecs really a very good solution for this kind of data? This is also the reason why I pointed out "it does not perform any explicit rate–distortion modeling or entropy analysis." Simply changes the QPs in standard codecs (as explained in your rebuttal) is not a good solution. It needs a comprehensive entropy model design to compress such a different kind of data, and it could be an ad-hoc or even end-to-end entropy model.
>
> In addition, my concern on the novelty of this method remains, so I still keep my score.

---

> > ### Author Response · Authors · 2025-11-27
> > **Comments from author**
> >
> > Thanks for your valuable comments.
> >
> > First, as we presented in both the paper and rebuttal, using a standard video codec is the requirement of the industry. MPEG released a very clear requirement document to highlight this problem. Therefore, using standard video codecs is not only proposed by our paper, while a requirement of the whole industry.
> >
> > Second, using the standard video codecs reflects that we cannot improve or modify the developed mature codecs, the first and most important is to generate a better PLAS image/video to fully exploit its performance, which motivates the proposed Inter-PLAS method.
> >
> > Third, for the entropy model, we not only explore using different QP, but also how to conduct PLAS and coordinate compression with video codec and LZMA. The strategy to use PLAS will influence the smoothness of the final PLAS image/videos, as well as influence the final entropy. Besides, discussing a complex or end-to-end entropy model requires that we have reached an agreement about 3D to 2D conversion. Currently, we are still in the stage that how to generate a better 2D PLAS image/videos.
> >
> > Please do not ignore the clear requirement from the industry, which is the most practical problem that we should emphasize for the application track.

---

### Official Review · Reviewer_xhFL · 2025-11-02

**Soundness:** 2
**Presentation:** 2
**Contribution:** 1
**Rating:** 4
**Confidence:** 2

**Summary:**

The paper proposes a new 3D Gaussian splat compression scheme, where a new variant of the Parallel Linear Assignment Sorting (PLAS) procedure is used to increase inter-frame similarities between mapped I- and P-frames, thus improving compression efficiency when a video-based codec is used.

**Strengths:**

Compression of 3D Gaussian splats is a relevant topic. The technical description of the proposal in section 4 is fairly easy to follow. Reuse of existing video / point cloud-based codec means standard-compliant and easy adaptation.

**Weaknesses:**

1. In the big picture of 3D Gaussian splats compression, given an existing video or point cloud-based codec is used downstream, the proposal to tweak PLAS (which the authors themselves admit is itself "heuristic" (pg.3)) in order to increase inter-frame similarities as a pre-processing step is a pretty minor hack that is not theoretically grounded. Specifically, in Section 4.1, the proposed spatial-stable initialization (SSI) is basically sorting via Morton code, which is long used in voxels when 3D point clouds voxelized into successively embedded cubes. This seems quite minor. Second, in Section 4.2, the anchor-based PLAS refinement amounts to "disabling the generation of the blurred target $T_i$ in PLAS and use $I_{PLAS}$ instead as the target to progressively smooth $P_{ini}$", which is also a minor tweak.

2. The description is not always clear, even for compression non-experts outside the Gaussian splats topic. What's meant by "end-to-end entropy supervision" What does "stable" mean in "stable I-frame PLAS image generation"? What does "anchor-based PLAS refinement" mean? There are also awkward / unnatural English expressions throughout the manuscript, such as "raising the upper limit of the quality" (abstract), "exhibits evidently improved performance" (abstract), "experiment results" rather than "experimental results" (abstract), "motivates commensurate attention" (intro).

**Questions:**

1. How would the I- and P-frame (and possibly B-frames) sequencing affect the proposal? For example, if P-frames $t$ can reference earlier frame $t-k$ for $k \geq 2$ for motion prediction in an IPPP frame sequence?

2. How should the blur parameters like $\sigma$, which control the speed and extent of the blurring processing, be optimized given the length of the generated IPPP sequence?

---

> ### Author Response · Authors · 2025-11-19
> **Response for weakness**
>
> # For Weakness 1:
>
> Thanks for your insightful comments. We agree that PLAS appears to be heuristic but, in our opinion, its potential in building a lightweight 3DGS compression framework should be well recognized. **More specifically, MPEG recently adopted PLAS in the current video-based GS compression anchor**, showing its potential in practical applications from an industrial perspective, as well as making PLAS part of a state-of-the-art (SOTA) method.
>
> Since we are still in an early stage of research on video-based GS compression methods, the vanilla PLAS is for a single GS frame and the inter-frame case is not supported, the current video-based GS compression anchor only considers the case where the tracked information is available. However, in most real applications, the tracked information does not exist. Therefore, **a lightweight solution to this problem is heavily weighted by both standardization organizations** such as MPEG and the industry, as it is essential for the wide adoption of 3DGS.
>
> Our work was partially motivated by noting that using the EMD to establish track information is rather time-consuming. The use of Morton code in the proposed SSI transforms the original primitive-to-primitive matching problem into a sorting problem, which is much easier to handle. Moreover, introducing the anchor-based PLAS further refines the established track information in a complexity-controllable and efficient way, enabled by its heuristic mechanism. These lightweight enhancements greatly improved the practicality of PLAS in the GS sequence compression.
>
> Another meaningful contribution of this work is the proposed pipeline compatible with SOTA video codecs, including VTM, HM, FFMPEG libx265 and libx264. The developed framework includes the methods for channel padding, YUV sequences generation and codec configuration. The validity of these modules has been verified using real-world 3DGS data, as reported in the experiment results. We also conducted a series of experiments to investigate how to better utilize existing video codecs. Considering that the adopted video codecs have already been implemented and supported by even mobile devices, the proposed compression pipeline provides a very promising pathway for the wide civilian deployment of 3DGS.
>
> # For Weakness 2:
>
> Thanks for your comments and questions. Here, “end-to-end entropy supervision” means using the RD loss to train a learning-based compressor. We have explained the meaning of “stable” in lines 180-183 on Page 4 of our manuscript with the help of Fig. 2, as well as in lines 201-206. It can be observed that given the same GS samples, performing the vanilla PLAS might produce totally different 2D maps. We refer to this property of the vanilla PLAS as being “unstable”. Our method addressed this issue, leading to more “stable” image generation where the randomness of the vanilla PLAS is mitigated.
>
> For other expressions you mentioned, we have carefully modified them and made them as easy to understand as possible for all readers. They are highlighted in blue in the revised manuscript and listed below for your reference:
>
> “Using an effective context model and rate-distortion (RD) loss function, they can achieve a compression ratio of more than 60 to 100x without noticeable distortion”
>
> “Even though we may fix the random seed, the permutation-invariant characteristic of GS data can still lead to different results, indicating current PLAS is non-deterministic and unstable. For GS sequence coding, a stable, simple, and effective initialization method is required for generating highly reproducible results.”
>
> “we disable the generation of the blurred target $T_i$ in PLAS and use $I_{PLAS}$ instead as the anchor to progressively smooth $P_{ini}^{'}$, resulting in the proposed anchor-based PLAS refinement.”
>
> “achieving higher compressibility while simultaneously providing a higher quality upper bound.”
>
> “Experimental results show that…”
>
> “which attracts considerable attention for effective GS data compression”

---

> ### Author Response · Authors · 2025-11-19
> **Response for question**
>
> # For Question 1:
>
> Thanks for your valuable questions. The order of P/B frames will not influence the results evidently. There do exist some motions for the rendered video from the GS sequence. But after converting GS into 2D maps using PLAS, this motion information is no longer identifiable because PLAS involves pixel-level optimization and permutation, and the original local semantic information is not preserved.
>
> When we generate the Inter-PLAS images, the I-frame is utilized as the reference and all the following frames are made close to it. The motion region will be reflected in terms of the different attributes on the same image index.
>
> We conduct the following new experiment to quantify the influence of compression sequencing on PLAS video. Specifically, we used HM software to compare two different compression configurations: “low_delay” and “randomaccess’’. For the “randomaccess” mode, frames 1-7 are B-frames with compression order: 0-4-2-1-3-6-5-7. For “low-delay”, the compression order is 0-1-2-3-4-5-6-7 with 1 to 7 as P-frames. We added a new section in Appendix A.11 to report the newly obtained RD curves. The finding is that using “randomaccess’’ reports only 3.63% performance gain over “low_delay’’.
>
> # For Question 2:
>
> Thanks for your valuable question. In fact, the blur parameters like sigma are all all-intra parameters. They have no relationship with sequence length. Thus, we guess you may want to ask about the effect of $\tau$, which controls the speed of inter-PLAS generation. Larger $\tau$ indicates using more iterations for P-frame PLAS image generation, which generally produces a closer image compared with I-frame images, as well as longer computation time. Theoretically, $\tau$ should be increased with the increase of the length of IPPP sequences. The reason is that with the increase of the sequence length, the difference between the P-frames and I-frames increases, which means the inter-PLAS requires more iterations to refine P-frames. We perform a new ablation study here. For P-frames of ``bartender'', we set: 1) D1: initial $\tau_0$ as 0.6, and $\tau_t = min (\tau_0 \times 1.1^{t-1}, 0.95), t>=1$, in which $\tau_t$ gradually increases to 0.95; and 2) D1: initial $\tau_0$ as 0.95, and $\tau_t = max (\tau_0 \times 0.9^{t-1}, 0.6), t>=1$, in which $\tau_t$ gradually decreases to 0.6.
>
> We can see that using a fixed $\tau$ of 0.95 reports the best performance, while the case with an increased $\tau$ introduces around 2.12% gain compared with the case that uses a decreased $\tau$. This corroborates our explanation that for the GS sequences, the P-frames distant from the I-frames exhibit greater differences and therefore should be assigned a larger $\tau$. We have added this new ablation study in Appendix A.12.

---

> ### Author Response · Authors · 2025-11-27
> **Looking forward to your rebuttal assessment**
>
> Dear Reviewer **xhFL**,
>
> Thanks for your time to review our paper. We have carefully addressed your comments and concerns, as presented in our detailed response and the revised manuscript.
>
> We are looking forward to your further assessment.
>
> Best,
>
> Authors

---

### Comment · Area_Chair_2gm3 · 2025-11-23

Dear reviewers,

Please review the rebuttal and discuss with the authors.

Thanks,
AC

---

### Author Response · Authors · 2025-11-25
**Looking forward to your new comments**

Dear Reviewers,

We would like to sincerely thank you for your time in reviewing our paper and effort to assess our work. We have carefully read your comments and revised our paper. In the rebuttal process, we have added 9 new experiments (Appendix A.11 – A.19) to address your comments and highlight our contribution and novelty.

We look forward to your further assessment.

Best regards,

Authors of No.529

---

### Meta-Review · Area_Chair_hn2C · 2026-01-07

**Summary:**

The reviewers' collective concerns primarily focused on the methodological novelty, empirical completeness, and the practical justification of the proposed framework. While reviewers xhFL and An3v noted the practical potential of utilizing standard video codecs for 3DGS sequence compression, wnCS and qDJS raised significant concerns regarding the technical depth and the actual competitive advantage of the method. A critical discussion emerged regarding whether the proposed Inter-PLAS is a sufficient advancement or merely a heuristic refinement. Furthermore, the effectiveness of using traditional video codecs was questioned, as PLAS-generated images deviate significantly from natural video statistics, leading to inconsistent performance gains across different bitrate regions. My suggested decision to reject is informed by the observation that the method's trade-offs—specifically the high post-processing overhead versus the achieved rate-distortion (RD) performance—do not currently present a compelling advantage over existing or emerging paradigms.

**Reviewer Concerns:**

While the rebuttal addressed several empirical gaps, such as providing additional baseline comparisons (Appendix A.17) and residual analysis (Appendix A.13), several core issues remain outstanding. A key concern, emphasized by the ongoing evaluation of the technical landscape, is that the high computational cost of the iterative PLAS process is increasingly difficult to justify. Given the emergence of feed-forward 3DGS generation methods (e.g., the VGGT family) that achieve near real-time performance, a method requiring heavy post-processing must demonstrate a much more decisive lead in RD performance to be competitive. However, the proposed method does not consistently outperform established anchors like GPCC across all test sequences and bitrates. Additionally, as wnCS pointed out, the lack of a fundamental innovation in entropy modeling leaves the framework as a combination of existing tools rather than a significant theoretical step forward. These limitations in performance consistency and methodological depth suggest that the paper, in its current form, does not yet meet the high bar for an ICLR acceptance.

**Reviewer Scores:**

I anticipate that the final consensus would lean toward rejection if the discussion were concluded now. Reviewer xhFL would likely move to a 6 due to the clarified stability and terminology. However, Reviewer qDJS would likely maintain or even lower their score, as the competitive analysis reveals that the method's RD gains do not sufficiently compensate for its lack of encoding speed, especially when compared to more efficient emerging approaches. Reviewer wnCS is expected to maintain their score, as the fundamental concern regarding the incremental nature of the novelty remains unaddressed. Reviewer An3v might maintain a 6. I find that the critical technical shortcomings highlighted by the other reviewers regarding the quality-complexity trade-off carry more weight in the final assessment. Consequently, without a more substantial improvement in both performance and efficiency, the paper does not reach the threshold for a positive recommendation.

---

### Decision · Program_Chairs · 2026-01-26

Reject